# Don't Overthink with Pixels: Efficient Reasoning for Segmentation

**Song Wang**[1 2]   **Gongfan Fang**[2]   **Lingdong Kong**[2]   **Xiangtai Li**[3]   **Jianyun Xu**[4 †]
**Sheng Yang**[4]   **Qiang Li**[4]   **Jianke Zhu**[1 5 *]   **Xinchao Wang**[2 *]

 **Project Page:** PixelThink.github.io

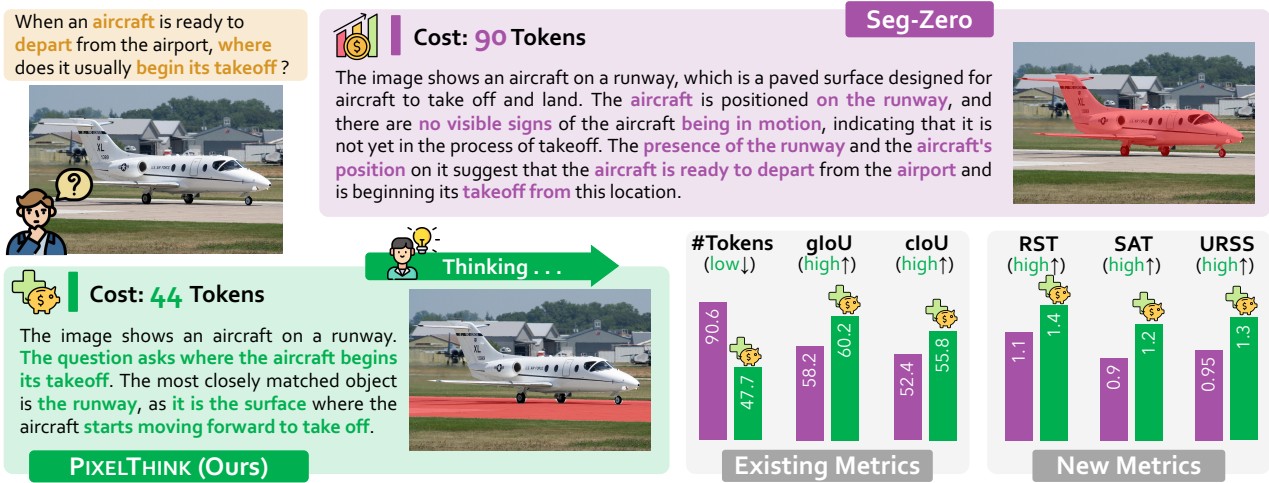

*Figure 1.* **Motivation of Efficient Reasoning for Segmentation (PIXELTHINK).** We propose a novel scheme for reasoning segmentation that effectively regulates reasoning length based on task *difficulty* and model *uncertainty*. Our method improves segmentation quality while significantly reducing token usage and inference latency. A suite of metrics is introduced for holistic evaluations of reasoning quality, segmentation accuracy, and computational efficiency.

## Abstract

Existing reasoning segmentation approaches typically fine-tune multimodal large language models (MLLMs) using image-text pairs and corresponding mask labels. While recent efforts leverage reinforcement fine-tuning to further enhance reasoning ability, they often suffer from overthinking and produce uniformly verbose reasoning chains irrespective of task complexity. To address this problem, we propose PIXELTHINK, a simple yet effective scheme that integrates externally estimated task difficulty and internally measured model uncertainty to regulate reasoning generation within a reinforcement learning paradigm. The model learns to compress reasoning length in accordance with scene complexity and predictive confidence. To support comprehensive evaluation, we introduce ReasonSeg-DIFF, an extended benchmark with annotated reasoning references and difficulty scores, along with a suite of metrics designed to assess segmentation accuracy, reasoning quality, and efficiency jointly. Experimental results demonstrate that the proposed approach not only improves segmentation performance but also significantly reduces inference latency by 30.4%, cutting token usage by 48.2%.

*Corresponding authors. †Project leader. [1]Zhejiang University [2]National University of Singapore [3]Nanyang Technological University [4]Unmanned Vehicle Dept., CaiNiao Inc., Alibaba Group [5]Shenzhen Loop Area Institute. Correspondence to: Jianke Zhu <jkzhu@zju.edu.cn>, Xinchao Wang <xinchao@nus.edu.sg>.

*Proceedings of the 43rd International Conference on Machine Learning*, Seoul, South Korea. PMLR 306, 2026. Copyright 2026 by the author(s).

## 1. Introduction

Reasoning segmentation (Yu et al., 2016; Lai et al., 2024; Zhu et al., 2025b) is an emerging vision-language task that requires predicting pixel-level masks in response to complex natural language queries. In contrast to traditional semantic or instance segmentation (Chen et al., 2017; He et al.,

2017; Cheng et al., 2022), which depends on predefined class labels, reasoning segmentation involves grounding fine-grained referring expressions that encode attributes, spatial relations, or contextual information. This capability is critical for embodied tasks such as interactive robotics (Yin et al., 2023; Yu et al., 2025) and autonomous driving (Tian et al., 2024; Xie et al., 2025; Wang et al., 2025a; Hu et al., 2025). Advances in multimodal large language models (MLLMs) (Liu et al., 2023b; Wang et al., 2024a; Bai et al., 2025b) have facilitated the development of reasoning segmentation via supervised fine-tuning (SFT).

Representative approaches (Lai et al., 2024; Ren et al., 2024b; Zhang et al., 2024; Bai et al., 2024), such as the pioneering LISA (Lai et al., 2024), integrate pre-trained MLLMs with segmentation modules through additional vision-language supervision to enable language-guided segmentation. Despite achieving strong performance on in-domain tasks, these methods often face limitations in generalizing to out-of-distribution (OOD) scenarios, especially when presented with complex or ambiguous queries (Liu et al., 2025b; Shen et al., 2025). Moreover, the absence of explicit reasoning chains reduces interpretability and hinders effective error analysis.

To overcome the limitations of SFT-based methods, recent progress in LLM research (Chen et al., 2026; Li et al., 2025c) has motivated the adoption of reinforcement learning (RL) strategies to enhance reasoning capabilities and generalization. In particular, group relative policy optimization (GRPO) has shown strong performance in language domains such as mathematical and code reasoning without requiring additional supervision (Shao et al., 2024; Guo et al., 2025). Building on this, several works have extended GRPO to visual perception tasks (Team, 2025; Shen et al., 2025; Liu et al., 2025b;d; Feng et al., 2025b), achieving improved out-of-distribution generalization and generating explicit, interpretable reasoning paths. Nevertheless, these methods often suffer from overthinking and produce unnecessarily verbose reasoning chains in simple cases, which leads to increased computational cost and reduced efficiency (Feng et al., 2025a; Sui et al., 2025). As illustrated in Figure 1, Seg-Zero (Liu et al., 2025b) yields incorrect segmentation results, despite utilizing a redundant reasoning process involving *twice the number of tokens*. Moreover, the lack of standardized evaluation for reasoning quality hinders a thorough assessment of the benefits brought by explicit reasoning in segmentation.

In this paper, we propose PIXELTHINK for reasoning segmentation, which regulates reasoning length based on externally estimated task difficulty and internally measured model uncertainty. Each input is assigned a token budget based on its estimated difficulty and uncertainty, and GRPO is guided by soft length-aware rewards that gen-

tly penalize excessive reasoning, promoting conciseness when appropriate. To facilitate systematic evaluation, we introduce ReasonSeg-DIFF, an extended version of Reason-Seg (Lai et al., 2024), enriched with task difficulty annotations and dual-mode reasoning references (*short* and *long*). Our evaluation protocol jointly assesses segmentation accuracy (gIoU and cIoU) and reasoning score (RScore) using LLM-based ratings against reference reasoning chains. Additionally, we propose three efficiency-aware metrics to quantify the trade-off between segmentation performance and reasoning token usage. Specifically, RST and SAT assess the efficiency of reasoning and segmentation, respectively, while URSS provides a unified measure that comprehensively captures overall effectiveness across both aspects.

We conduct extensive experiments to benchmark PIXEL-THINK against state-of-the-art reasoning segmentation and efficiency methods (Chen et al., 2024b; Liu et al., 2025b; Aggarwal & Welleck, 2025). The results demonstrate that PIXELTHINK effectively regulates reasoning length in accordance with task difficulty, while maintaining reasoning quality and further enhancing segmentation accuracy. Additionally, we present exploratory analyses on the role of reasoning, including ablations under *no-thinking* conditions. These analyses confirm that necessary and concise reasoning improves segmentation performance, whereas excessive redundancy provides no additional benefit.

Our main contributions can be summarized as follows: 1) We propose a novel scheme PIXELTHINK that enables efficient reasoning segmentation by leveraging external task difficulty and internal model uncertainty to guide the reward process in reinforcement fine-tuning; 2) We build ReasonSeg-DIFF, a new benchmark with annotated reasoning references and difficulty scores, and establish a comprehensive evaluation protocol that covers reasoning quality, segmentation accuracy, and efficiency; 3) Extensive experiments validate the effectiveness of our approach in reducing reasoning length and enhancing segmentation performance. In-depth analyses under various reasoning strategies are also conducted to inform future research.

**Conflict of Interest Disclosure.** The authors J.X., S.Y., and Q.L. are employed by Alibaba Group, the organization responsible for developing the publicly available Qwen model series evaluated in this paper.

## 2. Related Work

**Reasoning Segmentation.** Referring expression segmentation (Kazemzadeh et al., 2014; Yu et al., 2016; Wu et al., 2024; Yang et al., 2024b; Chng et al., 2024; Huang et al., 2025; Zhang et al., 2025) extends traditional segmentation approaches (Ronneberger et al., 2015; Cheng et al., 2022; Kirillov et al., 2023; Xu et al., 2025a; Wang et al., 2025b)

to open-vocabulary settings by localizing objects described in natural language. Recent works leverage multimodal large language models (MLLMs) (Liu et al., 2023b; 2024; Wang et al., 2024a; Bai et al., 2025b) to integrate visual and linguistic reasoning, enabling flexible segmentation from free-form queries. LISA (Lai et al., 2024) introduces step-wise alignment between textual reasoning and object grounding. A series of subsequent works (Ren et al., 2024b; Bai et al., 2024; Zhu et al., 2025; Yang et al., 2023; Yuan et al., 2025) explore fine-tuning MLLMs with segmentation heads, leveraging token-level instructions for fine-grained prediction. Seg-Zero (Liu et al., 2025b) further improves generalizability through reinforcement fine-tuning (Shao et al., 2024), generating reasoning chains and reference tokens that guide segmentation modules. Despite these advances, current approaches still struggle with reasoning efficiency and adaptability to varying task difficulty levels. This work builds upon prior research and introduces an efficiency-aware reasoning scheme.

**Large Reasoning Models.** Large language models (LLMs) have demonstrated remarkable capabilities in multi-step reasoning through Chain-of-Thought (CoT) prompting (Wei et al., 2022; Zhang et al., 2023; Yu et al., 2023). Beyond prompting, recent efforts focus on optimizing the reasoning process via process reward models (Wang et al., 2024b; 2025c; Song et al., 2025), reinforcement fine-tuning (OpenAI, 2024; Team et al., 2025; Shao et al., 2024), and other test time scaling methods (Muennighoff et al., 2025; Liu et al., 2025a; Zuo et al., 2025). DeepSeek-R1 (Guo et al., 2025) employs group relative policy optimization (GRPO) (Shao et al., 2024) to elicit LLMs' latent reasoning capacity and achieves significant advances. Building on this paradigm, recent works have extended GRPO and LLM-based reasoning to the visual domain (Lab, 2025; Team, 2025; Liu et al., 2025d; Tan et al., 2025; Shen et al., 2025), enabling multimodal reasoning via vision-language models (Wang et al., 2024a; Bai et al., 2025b). However, current visual reasoning research lacks evaluation of both the reasoning process and perceptual outcomes. We bridge this gap by introducing a holistic evaluation protocol that jointly assesses reasoning quality and segmentation performance.

**Efficient Inference Methods.** Recent surveys (Feng et al., 2025a; Sui et al., 2025; Liu et al., 2025c; Qu et al., 2025) have highlighted the inefficiencies of current reasoning models, including excessive token usage and redundant reasoning steps. To mitigate these issues, a variety of strategies have been explored. TALE (Han et al., 2024) proposes allocating token budgets adaptively, while CoT-Valve (Ma et al., 2025b) employs model merging to train reasoning chains of different lengths via supervised fine-tuning. Reinforcement learning-based methods such as L1 (Aggarwal & Welleck, 2025) and O1-Pruner (Luo et al., 2025a) impose direct constraints on reasoning length during training. Fur-

ther improvements include draft-based generation (Xu et al., 2025b), skip mechanisms (Xia et al., 2025), pruning strategies (Hou et al., 2025) and self-training (Munkhbat et al., 2025). In the context of efficient MLLMs, efforts have primarily focused on adaptive input compression (Shang et al., 2024; Li et al., 2025b; Chen et al., 2024a; Yang et al., 2025; Xu et al., 2025c), while output-level reasoning optimization remains underexplored. We address this problem by introducing a reward-driven mechanism that regulates reasoning length based on task difficulty and model uncertainty.

## 3. Methodology

### 3.1. Overview

**Problem Definition.** Reasoning segmentation (Lai et al., 2024; Liu et al., 2025b) aims to generate accurate segmentation masks given an image $\mathcal{I}$ and a referring expression $\mathcal{E}$. In contrast to conventional referring segmentation, which directly maps inputs to segmentation outputs, our formulation additionally requires the model to produce an explicit intermediate reasoning process $\mathcal{R}$ to improve interpretability and generalization. The task thus involves generating both the reasoning chain $\mathcal{R}$ and the segmentation mask $\mathcal{M}$.

**Baseline.** We follow the standard setup (Liu et al., 2025b) as illustrated in Figure 2(a), where the overall framework consists of a reasoning model and a segmentation model. A multimodal large language model (MLLM), specifically Qwen2.5-VL (Bai et al., 2025b), is adopted as the reasoning backbone (Reason). With an image $\mathcal{I}$ and a referring expression $\mathcal{E}$ as inputs, the model generates two distinct outputs: $\mathcal{R}, \mathcal{S} = \text{Reason}(\mathcal{I}, \mathcal{E})$, where the reasoning chain $\mathcal{R}$ is a multi-step textual explanation that reflects the model's visual understanding and reasoning process. Segmentation reference tokens $\mathcal{S}$ are spatial priors including a bounding box and two points, which serve as inputs to the segmentation model. We utilize SAM series models (Kirillov et al., 2023; Ravi et al., 2024) as the segmentation module, which takes the segmentation reference token $\mathcal{S}$ predicted by the reasoning model as input and produces the final binary mask $\mathcal{M}$. This modular design enables a clear decoupling of reasoning and fine-grained segmentation.

**Reinforcement Fine-Tuning.** We adopt reinforcement fine-tuning (RFT) to explicitly regulate output characteristics by optimizing non-differentiable objectives through task-specific reward signals. As shown in Figure 2(b), our reward formulation integrates task difficulty and model uncertainty, promoting a balanced trade-off between reasoning quality and computational efficiency.

### 3.2. Difficulty and Uncertainty Estimation

**Task Difficulty.** To achieve efficient reasoning during training, we estimate an instance-level difficulty score

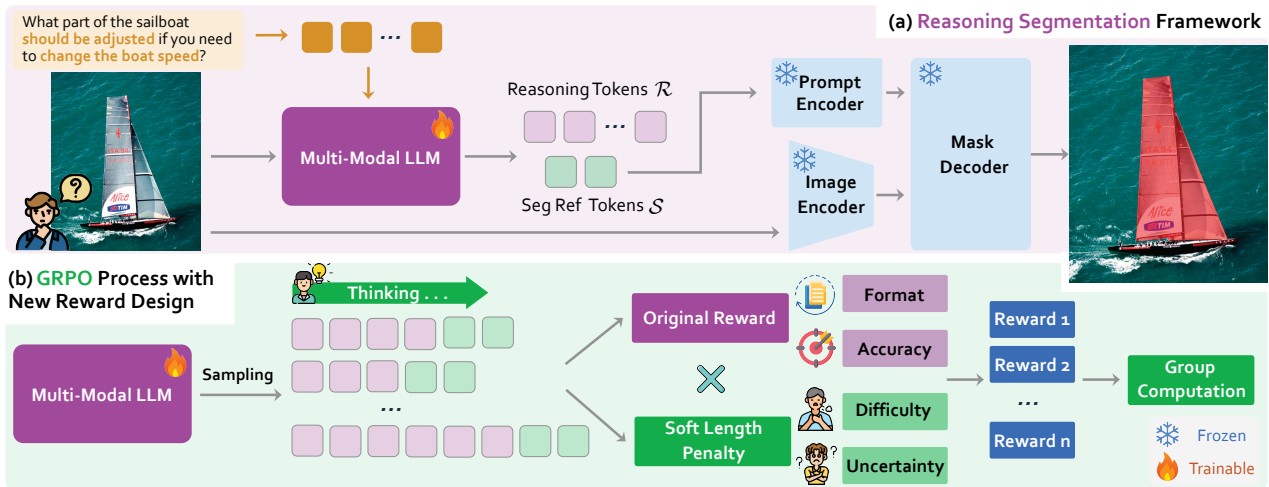

*Figure 2.* **Overview of PIXELTHINK.** (a) Workflow of the reasoning segmentation framework. Given an input image and query, the model generates a reasoning chain and segmentation reference that guides the segmentation outcome. The "snowflake" icon indicates frozen parameters, while the "fire" icon denotes modules undergoing active fine-tuning. (b) The group relative policy optimization (GRPO) procedure employed during reinforcement fine-tuning. Our new reward design incorporates both task difficulty and model uncertainty, enabling the model to learn efficient reasoning strategies.

$\mathcal{D} \in [1, 10]$ for each sample. Following the same process in benchmark construction (Section 4.1), we prompt a large MLLM to assess difficulty across three aspects: *scene complexity*, *segmentation challenge*, and *linguistic ambiguity*, and compute the final score as their average. These difficulty priors are then used to modulate token budget allocation in reinforcement fine-tuning.

**Model Uncertainty.** We also quantify model-internal uncertainty based on token-level confidence in the generated reasoning sequence. For each token, we compute the gap between the highest and second-highest predicted probabilities (Jiang & Gupta, 2019; Wang & Zhou, 2024), using this margin to estimate certainty. The overall uncertainty score $\mathcal{U}$ is defined as: $\mathcal{U} = 1 - \frac{1}{T} \sum_{t=1}^{T} (p_t^{(1)} - p_t^{(2)})$, where $p_t^{(1)}$ and $p_t^{(2)}$ are the top-2 probabilities at timestep $t$, and $T$ is the total number of tokens. A smaller margin indicates greater uncertainty, and the transformation ensures $\mathcal{U} \in [0, 1]$, with higher values corresponding to lower confidence. This internal self-assessment complements external task difficulty, jointly informing the adjustment of reasoning length.

### 3.3. Reward Design

**Original Reward.** We adopt the original reward design in Seg-Zero (Liu et al., 2025b), which captures both reasoning validity and segmentation accuracy. The reward function comprises the following components: $R_{\text{original}} = R_{\text{format}}^{\text{reason}} + R_{\text{format}}^{\text{seg}} + R_{\text{accuracy}}^{\text{seg}}$, which assesses reasoning format, segmentation format, and segmentation accuracy, respectively. $R_{\text{accuracy}}^{\text{seg}}$ comprises the evaluation of mask IoU, point-level, and bounding box-level L1 distance. The

original reward $R_{\text{original}}$ serves as the basis for subsequent reasoning length modulation.

**Soft Length Penalty.** To enable controllable reasoning length across tasks of varying complexity, we introduce a soft budget penalty that adaptively modulates the reward using both external task *difficulty* and internal model *uncertainty*. In contrast to prior approaches (Aggarwal & Welleck, 2025; Han et al., 2024), this mechanism encourages concise reasoning in simple scenarios while permitting elaboration in complex or ambiguous cases, thereby mitigating overthinking and balancing reasoning adequacy with efficiency. Given the difficulty score $\mathcal{D}$ and the uncertainty score $\mathcal{U}$, we define the expected reasoning token budget $L_{\text{budget}}$ as:

$$L_{\text{budget}} = \begin{cases} L_{\text{base}} + \alpha \cdot \mathcal{U}, & \text{if } \mathcal{D} \geq \tau_1 \\ L_{\text{low}}, & \text{if } \mathcal{D} < \tau_2 \\ \text{None}, & \text{otherwise} \end{cases} \quad (1)$$

where $\tau_1$ and $\tau_2$ divide tasks into *hard*, *medium*, and *easy* levels. $L_{\text{base}}$, $\alpha$, and $L_{\text{low}}$ are constants denoting the base budget for difficult tasks, the gain from uncertainty, and the minimal budget for easy tasks. We leave moderately difficult tasks unconstrained to allow learning flexibility in ambiguous regimes. The soft penalty is computed as:

$$s(L_{\text{used}}, L_{\text{budget}}) = \begin{cases} 1 - \beta \cdot (L_{\text{used}} - L_{\text{budget}}), \\ \quad \text{if } L_{\text{used}} > L_{\text{budget}}, \\ 1, \quad \text{otherwise.} \end{cases} \quad (2)$$

where $L_{\text{used}}$ is the word-level token count, and $\beta$ is a small penalty factor that ensures smooth reward decay without

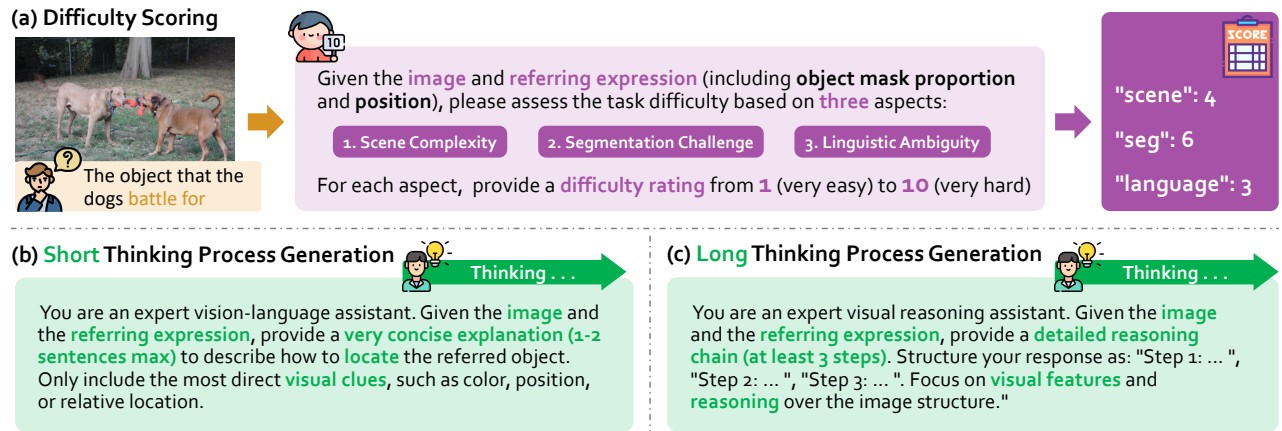

*Figure 3.* **The construction of ReasonSeg-DIFF.** (a) Design of the *Difficulty Scoring* scheme. (b) Generation of the *Short Thinking* process, characterized by concise and direct visual cues (1-2 sentences). (c) Generation of the *Long Thinking* process, featuring comprehensive and step-by-step reasoning. This dual-mode annotation enables adaptive evaluation across varying difficulty levels.

harsh clipping. This design maintains model stability during training and avoids discouraging minor budget exceedance that may improve output quality. The final reward is computed as: $R_{\text{final}} = R_{\text{original}} \cdot s(L_{\text{used}}, L_{\text{budget}})$. This formulation enables fine-grained adjustment of reasoning length, ensuring token usage aligns with *task complexity* and *model confidence* while avoiding hard constraints that may truncate informative reasoning during early training. Our empirical findings in Section 5.2 confirm that the chosen upper bounds are *sufficiently permissive* to allow learning while providing enough structure to suppress redundant output.

### 3.4. Reinforcement Fine-tuning Process

**Training with GRPO.** Following recent advances (Shao et al., 2024; Guo et al., 2025; Liu et al., 2025b), we adopt group relative policy optimization (GRPO) for reinforcement fine-tuning. The model is optimized to maximize the task-specific reward $R_{\text{final}}$ introduced in Section 3.3, which integrates reasoning quality, segmentation accuracy, and token efficiency into a unified objective. GRPO enhances training stability by comparing rewards within mini-batches at a group level, facilitating more consistent gradient updates and improving convergence with reduced variance. We present a detailed introduction of GRPO and SegZero in Section A.1 of the Appendix.

**Inference.** At inference time, the model operates under the same architecture and prompting schema as the baseline model. It generates both the reasoning chain $\mathcal{R}$ and segmentation mask $\mathcal{M}$ with image-text pairs as inputs. *No additional labels* or *reward feedback* are required during testing. This simple yet effective scheme facilitates seamless deployment and supports token-efficient, interpretable segmentation across diverse inputs.

## 4. Benchmark Construction & Evaluation Protocol

While existing reasoning segmentation benchmarks primarily focus on evaluating the final segmentation mask, they often overlook the reasoning quality and the efficiency of token usage. To address this gap and facilitate comprehensive evaluation, we construct ReasonSeg-DIFF, an extension of the ReasonSeg dataset (Lai et al., 2024) that includes task difficulty annotations and reference reasoning chains. This benchmark enables fine-grained assessment under varying levels of complexity.

### 4.1. The Construction of ReasonSeg-DIFF

As illustrated in Figure 3, the construction process consists of three parts, and we provide more details and examples with corresponding statistical information in Appendix B.

**Difficulty Scoring.** To quantify the intrinsic reasoning challenge of each sample, we propose a structured scoring framework based on three interpretable factors: (1) *Scene Complexity*, measuring the number and similarity of distractor objects; (2) *Segmentation Challenge*, capturing the spatial size, position, occlusion, and whether the target is a whole or a part; and (3) *Linguistic Ambiguity*, evaluating how explicit the referring expression is versus the need for visual inference. We categorize representative failure cases from the baseline (Liu et al., 2025b), finding that most are captured by our three primary dimensions (see Appendix C.3 for detailed statistics and examples).

As shown in Figure 3(a), we encode visual and textual priors into a unified prompt and query a large vision-language model (Qwen2.5-VL-72B (Bai et al., 2025b)) to independently score each aspect on a scale of [1–10], accompanied by a natural language explanation. The final difficulty score

is computed as the average of three dimensions, yielding a holistic and interpretable measure of instance-level complexity. To ensure reliability, this score is further cross-validated against human annotations, with details on annotation protocol and agreement statistics provided in Appendix B.2. We also categorize difficulty into three levels (*easy*, *medium*, and *hard*) using thresholds $\tau_1$ and $\tau_2$.

**Short and Long Thinking Process Generation.** To support evaluation under varying reasoning budgets, we construct two types of reference chains for each sample. The *short chain* conveys essential visual cues and confident identification in 1–2 sentences, while the *long chain* follows a structured multi-step format (*e.g.*, "Step 1... Step 2...") to emulate more comprehensive visual reasoning. Both chains are generated by prompting Qwen2.5-VL-72B with task-specific instructions and paired vision-text inputs as illustrated in Figure 3(b)(c). These only serve as evaluation references for assessing the informativeness and efficiency of model-generated reasoning.

### 4.2. Evaluation Protocol

With the constructed ReasonSeg-DIFF, we further introduce a comprehensive evaluation protocol for reasoning segmentation that jointly measures *segmentation accuracy*, *reasoning quality*, and *computational efficiency*. The protocol consists of three components detailed below.

**Segmentation Evaluation.** Following established settings (Yu et al., 2016; Kazemzadeh et al., 2014), we adopt two standard metrics: gIoU, the mean Intersection-over-Union (IoU) across all test samples, and cIoU, the cumulative IoU computed as the total intersection divided by the total union over the dataset. To incorporate efficiency into segmentation assessment, we propose **Segmentation Accuracy per Token** (SAT) inspired by prior research (Ma et al., 2025b), formulated as: $\text{SAT} = \frac{100 \times \text{gIoU}}{P \times \sqrt{T_{\text{num}}+1}}$, where $P$ denotes the number of model parameters (in billions), and $T_{\text{num}}$ is the average token length of the generated reasoning chains. The *square root* in the denominator provides a soft penalty for longer outputs, discouraging unnecessarily verbose reasoning without overly punishing small increases in token length (see Appendix C.1 for ablation). SAT favors models that achieve high segmentation accuracy with minimal reasoning overhead and compact model size.

**Reasoning Quality Evaluation.** To assess the quality of generated reasoning chains, we employ a large language model (LLM) (Yang et al., 2024a) to score predictions against the reference annotations introduced in Section 4.1. The evaluation covers three key dimensions: (1) *Completeness* — whether all necessary steps and information are included; (2) *Object Grounding* – the degree of alignment with the referred visual object; and (3) *Fluency and Clarity* – coherence, grammaticality, and readability.

Each aspect is rated on a 1–10 scale, and their average constitutes the overall reasoning score, denoted as RScore. To reflect reasoning efficiency, we further introduce **Reasoning Score per Token** (RST) as: $\text{RST} = \frac{10 \times \text{RScore}}{P \times \sqrt{T_{\text{num}}+1}}$. Following our difficulty-aware setting, RScore is computed using the *short* chain for *easy* and *medium* samples, and the *long* chain for *hard* ones. This evaluation protocol favors models that generate concise yet semantically rich reasoning, especially with limited tokens.

**Unified Metric.** To holistically evaluate reasoning segmentation across accuracy, reasoning quality, and computational efficiency, we propose the **Unified Reasoning Segmentation Score** (URSS): $\text{URSS} = (1 - \gamma) \times \text{RST} + \gamma \times \text{SAT}$, where $\gamma \in [0, 1]$ governs the relative emphasis on segmentation accuracy (SAT) and reasoning quality (RST). We set $\gamma = 0.7$ by default to reflect the *greater importance* of segmentation performance in practical applications.

## 5. Experiments

**Datasets.** We train exclusively on $9,000$ samples from RefCOCOg (Yu et al., 2016) *without any reasoning data*, following the same split as Seg-Zero (Liu et al., 2025b) for fair comparison. Evaluation is primarily conducted on ReasonSeg-DIFF derived from ReasonSeg (Lai et al., 2024), enabling *zero-shot* assessment across varying task complexities. ReasonSeg-DIFF includes 199 validation samples and 769 test samples, stratified by difficulty into 51/102/45 and 171/411/187 for easy, medium, and hard levels, respectively. We further report results on RefCOCO, RefCOCO+, and RefCOCOg to validate the common in-domain performance.

**Implementation Details.** We mainly initialize the reasoning model with Qwen2.5-VL-7B (Bai et al., 2025b) and adopt SAM2-Large (Ravi et al., 2024) as the segmentation backbone. Reinforcement fine-tuning is performed using the GRPO (Shao et al., 2024), with a KL divergence coefficient of $5 \times 10^{-3}$ and 8 samples per update. The initial learning rate is set to $1 \times 10^{-6}$. The soft length penalty parameters are set as follows: $L_{\text{base}} = 256$, $\alpha = 25$, $L_{\text{low}} = 96$, and $\beta = 2 \times 10^{-3}$. The thresholds $\tau_1$ and $\tau_2$ are set to 5.0 and 3.5, respectively. All experiments are conducted on 8 NVIDIA A100 GPUs with DeepSpeed (Rasley et al., 2020).

### 5.1. Main Results

**Quantitative Results on ReasonSeg-DIFF.** We evaluate our method on our ReasonSeg-DIFF benchmark, which incorporates fine-grained difficulty annotations and reference reasoning chains to facilitate comprehensive assessment across reasoning quality, segmentation accuracy, and efficiency. As Seg-Zero (Liu et al., 2025b) is the only existing approach that jointly provides explicit reasoning chains and segmentation masks, we adopt it as the primary baseline

*Table 1.* Overall evaluation results on the proposed ReasonSeg-DIFF benchmark.

| Method | Reasoning Quality | | | Segmentation Performance | | | Overall |
|---|---|---|---|---|---|---|---|
| | #Token↓ | RScore↑ | RST↑ | gIoU(%)↑ | cIoU(%)↑ | SAT↑ | URSS↑ |
| *Source of ReasonSeg Data: Validation Set* | | | | | | | |
| ○ Seg-Zero (2025b) | 90.79 | 7.67 | 1.14 | 61.63 | 52.56 | 0.92 | 0.99 |
| ○ Seg-Zero (2025b) + Prompt | 57.62 | 7.31 | 1.36 | 62.50 | 59.22 | 1.17 | 1.23 |
| ○ Seg-Zero (2025b) + L1-Exact (2025) | 65.66 | 5.73 | 1.00 | 40.97 | 42.47 | 0.72 | 0.80 |
| ○ Seg-Zero (2025b) + L1-Max (2025) | 61.21 | 4.37 | 0.79 | 61.75 | 57.39 | 1.12 | 1.02 |
| ● **PIXELTHINK** (Ours) | **46.98** | 6.92 | **1.43** | **63.81** | **62.69** | **1.32** | **1.35** |
| *Source of ReasonSeg Data: Test Set* | | | | | | | |
| ○ Seg-Zero (2025b) | 90.58 | 7.67 | 1.14 | 58.20 | 52.37 | 0.87 | 0.95 |
| ○ Seg-Zero (2025b) + Prompt | 57.84 | 7.33 | 1.37 | 58.15 | 53.45 | 1.08 | 1.17 |
| ○ Seg-Zero (2025b) + L1-Exact (2025) | 65.24 | 5.71 | 1.00 | 39.63 | 34.42 | 0.70 | 0.79 |
| ○ Seg-Zero (2025b) + L1-Max (2025) | 61.61 | 4.31 | 0.78 | 58.20 | 47.44 | 1.05 | 0.97 |
| ● **PIXELTHINK** (Ours) | **47.66** | 6.92 | **1.42** | **60.17** | **55.77** | **1.23** | **1.29** |

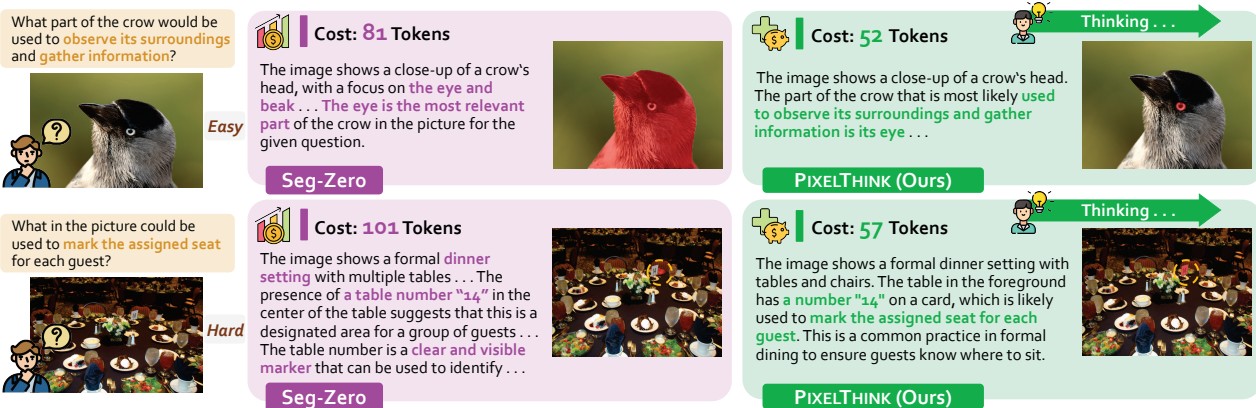

*Figure 4.* **Qualitative comparisons** between Seg-Zero (Liu et al., 2025b) and the proposed PIXELTHINK. Additional visual examples are shown in Figure B of the Appendix. We select representative samples across different difficulty levels.

for comparison. To examine efficient reasoning generation, we additionally compare with three length-aware baselines: (1) **Prompt**, which imposes a token limit through prompt-level constraints; (2) **L1-Exact**, and (3) **L1-Max**, two reinforcement fine-tuning strategies from L1 (Aggarwal & Welleck, 2025) that incorporate different reward formulations to regulate output length for LLM reasoning. To ensure fair comparison with Seg-Zero, all methods are fine-tuned exclusively on the RefCOCOg (Yu et al., 2016) training split without any additional annotations.

Table 1 presents the overall results on the validation and test sets of ReasonSeg-DIFF. Simple prompt-level constrains that produce shorter, more concise reasoning chains consistently improve segmentation performance, supporting our claim that *efficient reasoning leads to more accurate masks*. Moreover, Our method achieves substantial reductions in reasoning token usage while simultaneously enhancing segmentation accuracy. Unlike prompt-based and L1-style baselines that enforce rigid length constraints, our

approach preserves reasoning quality, resulting in a *more favorable balance* between performance and efficiency. Additionally, Table E in the Appendix reports difficulty-level breakdowns, where our method consistently surpasses all baselines across easy, medium, and hard subsets.

**Quantitative Results on Existing Benchmark.** We further evaluate the generalization capability of our method on four widely-used benchmarks: ReasonSeg (Lai et al., 2024) and the standard referring expression segmentation datasets RefCOCO, RefCOCO+, and RefCOCOg (Yu et al., 2016). On ReasonSeg, we compare against state-of-the-art approaches including OVSeg (Liang et al., 2023), ReLA (Liu et al., 2023a), LISA (Lai et al., 2024), Grounded-SAM (Ren et al., 2024a), and SAM4MLLM (Chen et al., 2024b). As reported in Table 2(a), our method achieves the highest segmentation accuracy while using significantly fewer reasoning tokens. Notably, our method with Qwen2.5-VL-3B model performs on par with prior 7B counterparts, highlighting the efficiency of PIXELTHINK. We also report results on RefCOCO, Ref-

*Table 2.* Performance comparison on existing benchmarks. Symbol [†] denotes scores reported in the paper while [*] denotes our reproduction with official code and model checkpoint.

(a) Reasoning segmentation on ReasonSeg.

| Method | val | | test | |
|---|---|---|---|---|
| | gIoU | cIoU | gIoU | cIoU |
| OVSeg (2023) | 28.5 | 18.6 | 26.1 | 20.8 |
| ReLA (2023a) | 22.4 | 19.9 | 21.3 | 22.0 |
| Grounded-SAM (2024a) | 26.0 | 14.5 | 21.3 | 16.4 |
| LISA-7B-LLaVA1.5 (2024) | 53.6 | 52.3 | 48.7 | 48.8 |
| LISA-13B-LLaVA1.5 (2024) | 57.7 | 60.3 | 53.8 | 50.8 |
| SAM4MLLM (2024b) | 46.7 | 48.1 | - | - |
| Qwen2.5VL-3B (2025b) + SAM2 | 53.8 | 44.1 | 47.6 | 37.4 |
| Seg-Zero-3B[†] (2025b) | 62.6 | 58.5 | 56.1 | 48.6 |
| Seg-Zero-7B[†] (2025b) | 62.6 | 62.0 | 57.5 | 52.0 |
| Seg-Zero-3B[*] (2025b) | 59.1 | 48.8 | 52.5 | 43.4 |
| Seg-Zero-7B[*] (2025b) | 61.6 | 52.6 | 58.2 | 52.4 |
| PIXELTHINK-3B | 62.3 | 58.5 | 58.8 | 52.1 |
| **PIXELTHINK-7B** | **63.8** | **62.7** | **60.2** | **55.8** |

(b) Referring expression segmentation (cIoU) on RefCOCO (+/g).

| Method | RefCOCO | RefCOCO+ | RefCOCOg |
|---|---|---|---|
| | testA | testA | test |
| LAVT (2022) | 75.8 | 68.4 | 62.1 |
| ReLA (2023a) | 76.5 | 71.0 | 66.0 |
| LISA-7B (2024) | 76.5 | 67.4 | 68.5 |
| PixelLM-7B (2024b) | 76.5 | 71.7 | 70.5 |
| MagNet (2024) | 78.3 | 73.6 | 69.3 |
| PerceptionGPT-7B (2024) | 78.6 | 73.9 | 71.7 |
| Seg-Zero-3B[†] (2025b) | 79.3 | 73.7 | 71.5 |
| Seg-Zero-7B[†] (2025b) | 80.3 | 76.2 | 72.6 |
| Seg-Zero-3B[*] (2025b) | 76.0 | 70.6 | 68.8 |
| Seg-Zero-7B[*] (2025b) | **79.4** | 73.7 | 73.2 |
| PIXELTHINK-3B | 78.7 | 72.9 | 72.2 |
| **PIXELTHINK-7B** | 79.3 | **74.8** | **73.9** |

COCO+, and RefCOCOg using their standard test splits. As shown in Table 2(b), our method delivers competitive performance across all datasets, confirming its generalization in both in-domain and out-of-domain settings.

**Qualitative Results.** Figure 4 presents visual comparisons between our method and Seg-Zero across various scenes. PIXELTHINK consistently predicts more precise segmentation masks while generating substantially shorter reasoning chains. In contrast, Seg-Zero frequently exhibits *overthinking*, producing lengthy and redundant explanations that fail to improve segmentation quality.

## 5.2. Diagnostic Experiments

We conduct diagnostic experiments on the validation set of ReasonSeg-DIFF for further exploration. Additional results and implementation details are provided in the Appendix C.

**Overall Inference Latency and Bottleneck Analysis.** To verify the practical efficiency gains, we measure the wall-clock inference time and throughput. As shown in Table 3, PIXELTHINK achieves a 30.4% reduction in overall latency

*Table 3.* Overall inference latency and time breakdown analysis (ms) using a single NVIDIA A100 GPU.

| Method | Overall Latency ↓ | MLLM | SAM2 | Others |
|---|---|---|---|---|
| Seg-Zero (2025b) | 1282.48 | 1023.81 | 66.46 | 192.21 |
| **PIXELTHINK** | **892.93** | **642.56** | 67.69 | 182.68 |
| *Improvement* | −30.4% | −37.2% | - | - |

*Table 4.* Ablation of our PIXELTHINK scheme.

| Difficulty | Uncertainty | #Token ↓ | gIoU(%)↑ | cIoU(%)↑ |
|---|---|---|---|---|
| | | 86.92 | 59.65 | 50.92 |
| ✓ | | **39.95** | 62.07 | 58.35 |
| | ✓ | 42.41 | 59.54 | 52.37 |
| ✓ | ✓ | 46.98 | **63.81** | **62.69** |

(1282ms → 893ms) compared to the baseline (Liu et al., 2025b). The results reveal that in the baseline, MLLM reasoning (1023ms) accounts for 79.8% of the total inference time, identifying it as the dominant bottleneck compared to the segmentation module (SAM2, 66ms). By efficiently compressing the reasoning chain, our method reduces this bottleneck by 37.2% (381ms), effectively translating token savings into significant real-world acceleration without compromising segmentation accuracy.

**Ablation on PIXELTHINK Scheme.** We ablate the two central components of PIXELTHINK, *task difficulty* and *model uncertainty*, to analyze their individual and joint contributions. For uncertainty-guided control only, we also divide the uncertainty scores into three levels and assign token budgets accordingly. As shown in Table 4, incorporating either difficulty or uncertainty alone reduces reasoning length and yields improvements in segmentation accuracy. Their combination achieves the optimal performance, confirming the complementary nature of the two signals for effective and efficient reasoning.

**Ablation on Difficulty Splits.** We further investigate the impact of difficulty granularity by varying the number of difficulty levels. In the 2-level setting, medium and hard samples are merged and assigned a uniformly larger token budget. In the 4-level setting, the medium category is further subdivided, with finer-grained length constraints applied. As illustrated in Table 5, the 3-level split offers the best balance between segmentation accuracy and reasoning efficiency.

**Ablation on Token Budget Allocation.** We next ablate the token budget configuration used for the soft length penalty. Specifically, we set the base upper bounds to $L_{base} = 256$ and $L_{low} = 96$, and assess alternative parameter configurations. As shown in Table 6, our approach consistently achieves substantial reductions in reasoning token usage while improving segmentation performance, demonstrating the robustness of the proposed budget design.

**No-thinking Mode Analyses.** Inspired by recent investi-

*Table 5.* Ablation of the difficulty splits in PIXELTHINK.

| Difficulty | #Tok↓ | gIoU↑ | cIoU↑ |
|---|---|---|---|
| - | 86.92 | 59.65 | 50.92 |
| 2 | 61.78 | **64.38** | 61.80 |
| 3 | **46.98** | 63.81 | **62.69** |
| 4 | 76.89 | 62.28 | 55.78 |

*Table 6.* Ablation on the token budget allocation details.

| Budget | #Tok↓ | gIoU↑ | cIoU↑ |
|---|---|---|---|
| - | 86.92 | 59.65 | 50.92 |
| (64, 256) | **24.71** | 62.12 | 59.57 |
| (96, 256) | 46.98 | **63.81** | **62.69** |
| (96, 384) | 60.18 | 60.18 | 53.29 |

*Table 7.* Ablation on the different no-thinking mode analyses.

| Method | #Tok↓ | gIoU↑ | cIoU↑ |
|---|---|---|---|
| No-thinking-RL | 0.00 | 60.19 | 49.49 |
| No-thinking-Prompt | 0.00 | 60.37 | 51.47 |
| Seg-Zero | 90.79 | 61.63 | 52.56 |
| **PIXELTHINK**(ours) | 46.98 | **63.81** | **62.69** |

gations into no-thinking paradigms for reasoning (Li et al., 2025a; Ma et al., 2025a), we extend this line of analysis to the pixel-level task. We implement two variants within the reinforcement learning framework: No-thinking-RL and No-thinking-Prompt. The corresponding details are provided in Appendix A.3. As demonstrated in Table 7, incorporating appropriate reasoning steps consistently improves segmentation accuracy, highlighting the benefit of efficient reasoning over naive or omitted inference.

## 6. Conclusion

In this paper, we propose PIXELTHINK, an efficiency-aware reasoning scheme for segmentation that explicitly regulates reasoning length based on task difficulty and model uncertainty. By introducing a soft length penalty and reward modulation, our method enables efficient chain-of-pixel reasoning and improves segmentation accuracy. Furthermore, we construct a difficulty-aware benchmark ReasonSeg-DIFF, and design holistic metrics that jointly assess reasoning quality, segmentation precision, and efficiency. Extensive experiments demonstrate that PIXELTHINK produces concise yet informative reasoning chains and consistently outperforms baselines across varying difficulty levels.

## Acknowledgments

This work is supported in part by the National Natural Science Foundation of China under Grant No. 62376244, in part by the Zhejiang Provincial Natural Science Foundation of China under Grant No. LD24F030001, by the Information Technology Center and State Key Lab of CAD&CG, Zhejiang University, and by the Ministry of Education, Singapore, under the Academic Research Fund Tier 1 (FY2026).

## Impact Statement

Our study does not involve human subjects, private user data, or sensitive personal information. The ReasonSeg-DIFF benchmark is constructed by reprocessing existing public datasets, with additional annotations generated using multimodal large language models and verified through controlled human annotation protocols. All annotators were briefed on the labeling task and did not provide person-ally identifiable information. We have carefully considered risks of potential dataset bias or fairness concerns: ambiguous linguistic expressions and visually cluttered scenes are explicitly analyzed in Appendix C.3, and limitations are discussed in the Appendix E.1. The code and annotations will be released under an open license to promote transparency, reproducibility, and ethical use. Our methodology does not provide direct means of harmful application, but we caution that efficiency-aware reasoning techniques could be misused for surveillance tasks if combined with sensitive data. We emphasize that our contributions are intended for academic research on multimodal reasoning and segmentation, with safeguards through open release policies and documentation. Further discussions are presented in Appendix E.

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

# Appendix

## A. Additional Implementation Details

In this section, we begin with a background overview of GRPO and Seg-Zero, and subsequently provide supplementary implementation details for the baseline models and the No-thinking methods presented in the main paper.

### A.1. Background on GRPO and Seg-Zero

**Group Relative Policy Optimization.** Group relative policy optimization (GRPO) (Shao et al., 2024) is a reinforcement learning algorithm designed to stabilize training in math reasoning tasks. Instead of computing rewards based on a separate value model like PPO (Schulman et al., 2017), GRPO evaluates groups of trajectories and assigns relative advantages. Within each group, the model identifies trajectories with higher performance (*e.g.*, better accuracy or lower redundancy) and encourages policies that increase their likelihood. This relative comparison mitigates issues of reward sparsity and scale mismatch, leading to more stable updates compared to standard PPO-style optimization. Formally, given a group of $G$ outputs $o_1, o_2, \ldots, o_G$ with corresponding rewards $r_1, \ldots, r_k$ from the old policy $\pi_{\theta_{\text{old}}}$, the relative advantage for output $i$ is defined as: $A_i = \frac{r_i - \mu_r}{\sigma_r}$, where $\mu_r$ and $\sigma_r$ represent the mean and standard deviation of the rewards within the group, respectively. The policy is optimized to maximize the expectation of $A_i$ under clipped likelihood ratios and a KL penalty term, similar to PPO:

$$
\begin{aligned}
\mathcal{J}_{GRPO}(\theta) = \mathbb{E}&\left[ q \sim P(Q), \{o_i\}_{i=1}^{G} \sim \pi_{\theta_{\text{old}}}(O \mid q) \right] \\
& \frac{1}{G} \sum_{i=1}^{G} \frac{1}{|o_i|} \sum_{t=1}^{|o_i|} \left\{ \min\left[ \frac{\pi_\theta(o_{i,t} \mid q, o_{i,<t})}{\pi_{\theta_{\text{old}}}(o_{i,t} \mid q, o_{i,<t})} \hat{A}_{i,t}, \right.\right. \\
& \left.\left. \text{clip}\left( \frac{\pi_\theta(o_{i,t} \mid q, o_{i,<t})}{\pi_{\theta_{\text{old}}}(o_{i,t} \mid q, o_{i,<t})}, 1 - \varepsilon, 1 + \varepsilon \right) \hat{A}_{i,t} \right] - \beta \, \mathbb{D}_{KL}[\pi_\theta \| \pi_{ref}] \right\}.
\end{aligned}
\tag{3}
$$

**About Seg-Zero Framework.** Seg-Zero (Liu et al., 2025b) is a recent framework that couples natural language reasoning with pixel-level segmentation. Instead of directly predicting a mask from the inputs, Seg-Zero generates an explicit reasoning chain that describes intermediate steps (*e.g.*, spatial relations, attributes, context) and grounding information, and then conditions a segmentation model (Ravi et al., 2024) to produce the final mask. This two-stage process improves interpretability and allows diagnostic analysis of where failures occur. Seg-Zero inherits the philosophy of DeepSeek-R1-Zero (Guo et al., 2025) by directly employing GRPO for post-training multimodal large language models. The reasoning process is conducted without any cold-start data or supervised guidance, which yields certain segmentation reasoning capabilities. However, it also introduces redundancy in the reasoning procedure, a central issue addressed in this work.

### A.2. Implementation Details on Baseline

**Seg-Zero Re-implementation.** To ensure a fair comparison, the evaluation of Seg-Zero (Liu et al., 2025b)'s 7B model is conducted using the official model checkpoint available in the public repository. Due to the unavailability of the 3B checkpoint, we re-train the model using the official codebase under the same experimental settings to ensure fair comparison. The prompts used for both models strictly adhere to the official implementation, as shown below. For consistency, the same prompt format is also adopted in our proposed PIXELTHINK.

---

**Original Prompt from Seg-Zero**

Please find "{Question}" with bbox and points.
Compare the differences between objects and find the most closely matched one.
Output the *thinking process* in <think> </think> and *final answer* in <answer> </answer> tags.
Output the *one bbox* and *points* of two largest inscribed circles inside the interested object in JSON format.
*i.e.*, <think> *thinking process here* </think>
<answer>"Bbox": [10, 100, 200, 210], "Point 1": [30, 110], "Point 2": [35, 180]</answer>

---

**Seg-Zero with Prompt for Short-Thinking.** For the prompt-based baseline, we utilize the official model checkpoint and explicitly *incorporate a token budget constraint into the prompt*, setting the upper limit to $64$ tokens during inference. The

specific prompt used is provided below:

---

**Short-thinking Prompt**

Please find "{Question}" with bbox and points.
Compare the difference between objects and find the most closely matched one.
Think step-by-step and explain your reasoning process in less than 64 tokens.
Output the *reasoning* in `<think>` `</think>` and the *final answer* in `<answer>` `</answer>` tags.
Output the *one bbox* and *points* of two largest inscribed circles inside the interested object in JSON format.
*i.e.*, `<think>` *short reasoning here* `</think>`
`<answer>`"Bbox": $[10, 100, 200, 210]$, "Point 1": $[30, 110]$, "Point 2": $[35, 180]$`</answer>`

---

**Adapt L1 for Reasoning Segmentation.** Since L1 (Aggarwal & Welleck, 2025) is designed to control the reasoning length of large language models (LLMs) (Guo et al., 2025; Yang et al., 2024a), its original implementation requires explicitly specifying the desired token count in the prompt. Besides, it is initially trained on the fixed-length DeepScaleR dataset (Luo et al., 2025b) using L1-Exact, followed by continued fine-tuning with L1-Max. To distinguish L1 from prompt-based baseline and ensure a fair comparison with Seg-Zero without relying on *additional reasoning data*, we adopt the same prompt format used in Seg-Zero for re-implementing L1 in the reasoning segmentation task. For both variants, we independently integrate the corresponding length control functions into the reward formulation. Specifically, L1-Exact uses a 64-token upper limit, while L1-Max allows up to 128 tokens, enabling comparable final reasoning lengths.

### A.3. Implementation Details on No-thinking Mode

For the implementation of No-thinking-Prompt, we directly use the official model checkpoint from Seg-Zero and apply a prompt that explicitly instructs the model not to produce any reasoning steps as below, thereby enforcing a "no-thinking" behavior. For No-thinking-RL, we follow the CLS-RL (Li et al., 2025a) framework by adopting a similar prompt format and removing the reasoning-format reward term from the GRPO reward function. We then re-train Seg-Zero using this modified reward formulation to obtain the final No-thinking-RL results.

---

**No-thinking Prompt**

Please find "{Question}" with bbox and points.
Compare the differences between objects and find the most closely matched one.
Output the final answer in the `<answer>` `</answer>` tag only.
The answer should include *one bbox* and the *points* of the two largest inscribed circles inside the interested object in JSON format, *i.e.*,
`<answer>`"Bbox": $[10, 100, 200, 210]$, "Point 1": $[30, 110]$, "Point 2": $[35, 180]$`</answer>`

---

## B. The ReasonSeg-DIFF Dataset

In this section, we present the dataset construction details, including the scoring prompts and representative examples from ReasonSeg-DIFF.

### B.1. Prompts for Scoring and Statistics

**Difficulty Scoring.** For both the RefCOCOg (Yu et al., 2016) training set and the validation/test splits constructed from ReasonSeg (Lai et al., 2024) to form ReasonSeg-DIFF, we need to assign a difficulty score for each sample. To achieve this, we generate *visual descriptions* based on mask properties (*e.g.*, size and position) and *textual descriptions* derived from the referring expressions (*e.g.*, expression length and the number of spatial terms). These descriptions are incorporated into the prompt. We then instruct Qwen2.5-VL-72B (Bai et al., 2025b) to rate each sample from three perspectives: (1) *Scene Complexity*, (2) *Segmentation Challenge*, and (3) *Linguistic Ambiguity*. The final difficulty score is computed as the average of these three ratings. The prompt used for this scoring process is provided below.

> **Difficulty Scoring Prompt**
>
> You are an expert in reasoning segmentation evaluation.
> Given the image and the referring expression: "{Question}", please assess the task difficulty based on the following three aspects:
> 1. *Scene Complexity*:- How many objects are visible in the scene?- How many of them are potentially related or visually similar to the target?
> 2. *Segmentation Challenge*:
> - What is the size and position of the target object?
> - Are there occlusions, overlaps, or visually similar objects nearby?
> - Is the mask describing the whole object or just a part? "{Visual Description}"
> 3. *Language Complexity*:
> - Does the referring expression explicitly point to the target object?
> - Or does it require additional reasoning to infer which object is referred to? "{Textual Description}"
> For each aspect, please provide a difficulty rating from 1 (very easy) to 10 (very hard), and summarize in the following Python dictionary format.
> *i.e.*, {"scene": 4, "segmentation": 6, "language": 3}

**Lable Statistics.** We further analyze the distribution of difficulty levels, namely *easy*, *medium*, and *hard*, as determined by our scoring framework for both RefCOCOg (Yu et al., 2016) and ReasonSeg (Lai et al., 2024). As summarized in Table A, RefCOCOg contains a significantly higher proportion of easy samples compared to ReasonSeg, which is consistent with commonly held expectations regarding the relative complexity of the two datasets. Notably, despite differences in data distribution between RefCOCOg and ReasonSeg, our method achieves superior *zero-shot* performance on ReasonSeg when trained solely on RefCOCOg.

*Table A.* Label statistics on difficulty distribution in training, validation and test set.

| Dataset | Easy | Medium | Hard |
|---|---|---|---|
| Training Set (RefCoCOg) | 3220 | 4810 | 970 |
| Validation Set (ReasonSeg) | 51 | 102 | 45 |
| Test Set (ReasonSeg) | 171 | 411 | 187 |

**Comparison with Direct MLLM Prompting.** To capture diverse sources of difficulty, we decompose the task into three dimensions: scene complexity, segmentation challenge, and linguistic ambiguity, reflecting variations in visual layout, object-level grounding, and textual clarity. We also compare our multi-dimensional decomposition with directly prompting a large MLLM (Bai et al., 2025b) to assign a single overall difficulty score. As shown in Table B, direct prompting yields a collapsed distribution, with the vast majority of samples classified as *easy*. In contrast, our decomposition produces a more balanced allocation across *easy*, *medium*, and *hard*, which is essential for reward shaping.

*Table B.* Difficulty label distribution under direct MLLM prompting versus our multi-dimensional decomposition.

| Method | Easy | Medium | Hard |
|---|---|---|---|
| Direct MLLM Prompting | **178** | 8 | 12 |
| Ours | 51 | **102** | 45 |

We also evaluated the downstream reasoning segmentation performance of models trained with above labels. As summarized in Table C, our approach leads to substantially higher gIoU and cIoU, demonstrating the effectiveness of fine-grained decomposition in providing diverse and informative supervision signals. These findings confirm that a coarse, single-score annotation is insufficient for difficulty-aware training, whereas our decomposition improves both the diversity of supervision and the resulting segmentation accuracy.

*Table C.* Downstream performance comparison between direct MLLM prompting and our decomposition.

| Method | #Token↓ | gIoU (%)↑ | cIoU (%)↑ |
|---|---|---|---|
| Training with MLLM Labels | **24.57** | 56.32 | 46.77 |
| Training with Ours | 46.98 | **63.81** | **62.69** |

**Reasoning Process Scoring.** For reasoning process evaluation, we adopt the following prompt and use reasoning chains from ReasonSeg-DIFF as reference annotations. The model-generated reasoning is evaluated by Qwen2.5-72B (Yang et al., 2024a) across three dimensions: *Completeness*, *Object grounding*, and *Fluency & Clarity*, providing a comprehensive assessment of reasoning quality.

---

**Reasoning Scoring Prompt**

You are an expert in evaluating reasoning quality for reasoning segmentation tasks.
Given the following predicted reasoning and reference reasoning, please score the prediction in three aspects from 1 to 10:
1. *Completeness*: Does it include all necessary steps and important information?
2. *Object Grounding*: Is it aligned with the referred object in the question?
3. *Fluency & Clarity*: Is the reasoning coherent, fluent, and grammatically correct?
The question is: "{Question}"
Reference Reasoning: "{Reference Text}"
Predicted Reasoning: "{Thinking Text}"
Return a Python dictionary with keys "completeness", "grounding", and "fluency".
*i.e.*, {"completeness": 8, "grounding": 7, "fluency": 9}

---

### B.2. Human Verification Details

To ensure the credibility of the automatically generated difficulty scores and reasoning chains in ReasonSeg-DIFF, we conduct a systematic human verification study on the validation set. Two independent annotators cross-checked a subset of 198 samples, evaluating task difficulty levels as well as reasoning chain quality across three criteria: linguistic fluency, logical coherence, and factual accuracy. Table D summarizes the agreement rates between human verification and the automatically generated labels. The results demonstrate that the large majority of automatically generated annotations are valid. Discrepancies mainly arise from borderline cases of scene complexity or ambiguous linguistic descriptions. These findings provide strong support for the credibility of the benchmark, while also motivating further refinement by expanding the proportion of human verification in future iterations.

*Table D.* Human verification results on the validation set (198 samples). Pass rates indicate the proportion of samples where the automatic labels and reasoning chains were confirmed by human annotators.

| Verification Criterion | Pass Count | Pass Rate (%) | Evaluation Standard |
|---|---|---|---|
| Task Difficulty Level | 176 | 88.9 | Easy / Medium / Hard consistency |
| Linguistic Fluency | 193 | 97.5 | Natural and comprehensible language |
| Logical Coherence | 186 | 93.9 | Reasoning steps are sequentially consistent |
| Factual Accuracy | 182 | 91.9 | Grounded and hallucination-free |

### B.3. Examples from ReasonSeg-DIFF

We present several representative examples from the constructed ReasonSeg-DIFF in Figure A, covering samples categorized as *easy*, *medium*, and *hard*. Each example includes its assigned difficulty scores along with the corresponding reference reasoning chains. For *easy* and *medium* cases, we recommend the use of *short* reasoning chains, whereas *longer* chains are preferable for *hard* samples. The annotation files are publicly available at our official project page for further reference.

### B.4. License

The ReasonSeg-DIFF dataset is released under the Attribution-ShareAlike 4.0 International (CC BY-SA 4.0)[1] license.

## C. Additional Experimental Results

In this section, we provide the detailed breakdown of our experimental results and extend the ablation analysis to complement the main paper.

**Detailed Difficulty-aware Evaluation.** We present the full results of our difficulty-aware evaluation on the ReasonSeg-DIFF test set in Table E. PIXELTHINK consistently surpasses all baseline methods across all difficulty levels. Notably, the

---

[1] https://creativecommons.org/licenses/by-sa/4.0/legalcode.

*Table E.* Difficulty-aware evaluation on the ReasonSeg-DIFF test set.

| Method | Reasoning Quality | | | Segmentation Performance | | | Overall |
|---|---|---|---|---|---|---|---|
| | #Token↓ | RScore↑ | RST↑ | gIoU(%)↑ | cIoU(%)↑ | SAT↑ | URSS↑ |
| *Difficulty Level: Easy* | | | | | | | |
| ○ Seg-Zero (2025b) | 84.97 | 8.07 | 1.24 | 68.65 | 65.85 | 1.06 | 1.11 |
| ○ Seg-Zero (2025b) + Prompt | 55.35 | 7.80 | 1.48 | 67.93 | 63.94 | 1.29 | 1.35 |
| ○ Seg-Zero (2025b) + L1-Exact (2025) | 68.03 | 6.42 | 1.10 | 51.95 | 43.77 | 0.89 | 0.96 |
| ○ Seg-Zero (2025b) + L1-Max (2025) | 60.15 | 4.73 | 0.86 | 68.40 | 62.15 | 1.25 | 1.13 |
| ● **PIXELTHINK** (Ours) | **44.73** | 7.56 | **1.60** | **70.25** | **67.49** | **1.48** | **1.52** |
| *Difficulty Level: Medium* | | | | | | | |
| ○ Seg-Zero (2025b) | 90.73 | 7.68 | 1.15 | 60.15 | 55.53 | 0.90 | 0.97 |
| ○ Seg-Zero (2025b) + Prompt | 58.37 | 7.30 | 1.35 | 59.89 | 56.15 | 1.11 | 1.18 |
| ○ Seg-Zero (2025b) + L1-Exact (2025) | 66.03 | 5.60 | 0.98 | 39.97 | 35.01 | 0.70 | 0.78 |
| ○ Seg-Zero (2025b) + L1-Max (2025) | 61.45 | 4.18 | 0.76 | 59.46 | 47.79 | 1.07 | 0.98 |
| ● **PIXELTHINK** (Ours) | **47.00** | 6.84 | **1.41** | **62.05** | **58.16** | **1.28** | **1.32** |
| *Difficulty Level: Hard* | | | | | | | |
| ○ Seg-Zero (2025b) | 95.37 | 7.26 | 1.06 | 44.37 | 28.93 | 0.65 | 0.77 |
| ○ Seg-Zero (2025b) + Prompt | 58.97 | 6.97 | **1.29** | 45.38 | 31.35 | 0.84 | 0.97 |
| ○ Seg-Zero (2025b) + L1-Exact (2025) | 60.94 | 5.29 | 0.96 | 27.61 | 18.94 | 0.50 | 0.64 |
| ○ Seg-Zero (2025b) + L1-Max (2025) | 63.28 | 4.22 | 0.75 | 46.09 | 30.36 | 0.82 | 0.80 |
| ● **PIXELTHINK** (Ours) | **51.79** | 6.50 | 1.28 | **46.80** | **35.05** | **0.92** | **1.03** |

*Table F.* Ablation on the uncertainty weight.

| Weight ($\alpha$) | #Token↓ | gIoU(%)↑ | cIoU(%)↑ |
|---|---|---|---|
| 0 | **39.95** | 62.07 | 58.35 |
| 25 | 46.98 | **63.81** | 62.69 |
| 35 | 71.10 | 62.66 | **63.13** |

*Table G.* Ablation on the length constrain for *medium* samples during training.

| *with* constrain | #Token↓ | gIoU(%)↑ | cIoU(%)↑ |
|---|---|---|---|
| ✓ | **42.10** | 60.92 | 53.84 |
| ✗ | 46.98 | **63.81** | **62.69** |

performance gains are most pronounced in the medium and hard categories, where our method effectively balances the necessary reasoning depth with token efficiency, mitigating the overthinking issues prevalent in Seg-Zero.

### C.1. Additional Ablation Results

All experiments in this section follow the main ablation setting, using the 7B model and evaluating on the validation split.

**Ablation on Uncertainty Weight.** We conduct an ablation study on the uncertainty ($\mathcal{U}$) weighting factor $\alpha$ for hard samples to evaluate its effect on reasoning length control and segmentation performance. As shown in Table F, selecting an appropriate weight is crucial for achieving optimal performance in both reasoning efficiency and segmentation accuracy.

**Ablation on Length Constrain for Medium Samples.** We also investigate the impact of applying length constrain to medium-difficulty samples. In the controlled variant, the reasoning length is limited to a maximum of 176 tokens. As shown in Table G, allowing medium cases to remain unconstrained provides greater flexibility in reasoning length, resulting in improved overall performance.

**Cross-model Validation.** To mitigate potential concerns about single-model bias from Qwen2.5-VL family (Bai et al., 2025b), we conduct cross-model validation by computing RScore with InternVL3-78B (Zhu et al., 2025a) in addition to Qwen2.5-VL-72B. As shown in Table H, both models yield consistent trends, demonstrating that the observed improvements are not limited to a single family of vision-language models. This further validates the robustness of our evaluation protocol.

*Table H.* Cross-model validation on ReasonSeg-DIFF. Results show consistent improvements across Qwen2.5-VL-72B and InternVL3-78B.

| Method | #Token↓ | RScore↑ | RScore (InternVL3)↑ |
|---|---|---|---|
| Seg-Zero+L1-Max | 61.21 | 4.37 | 4.01 |
| Ours | **46.98** | **6.92** | **6.63** |

*Table K.* Efficiency metric ablation. Square-root scaling in SAT yields more stable trade-offs for reasoning segmentation.

| Reasoning Length | #Token↓ | gIoU (%)↑ | SAT↑ | SAT (*w/o* $\sqrt{\cdot}$)↑ (Ma et al., 2025b) |
|---|---|---|---|---|
| Less Tokens | 36.98 (-10) | – | 1.48 (+12%) | 0.24 (+26%) |
| Original Tokens | 46.98 | 63.81 | 1.32 | 0.19 |
| More Tokens | 56.98 (+10) | – | 1.20 (-9%) | 0.16 (-16%) |

**About the Uncertainty Estimation.** We also compare entropy-based uncertainty estimation (Namdari & Li, 2019) with the adopted Top-2 margin strategy (Jiang & Gupta, 2019; Wang & Zhou, 2024). As summarized in Table I, entropy as uncertainty tends to generate longer reasoning chains and exhibits degraded segmentation accuracy, likely due to sensitivity to distributional noise in tail probabilities. By contrast, the Top-2 margin produces shorter chains while improving both gIoU and cIoU.

*Table I.* Comparison of uncertainty metrics. Top-2 margin yields shorter and more accurate results.

| Uncertainty Metric | #Token↓ | gIoU (%)↑ | cIoU (%)↑ |
|---|---|---|---|
| Entropy | 59.21 | 62.43 | 58.96 |
| Top-2 Margin (Ours) | **46.98** | **63.81** | **62.69** |

**About the Joint and Modular Decoder.** To explore end-to-end optimization, we train a lightweight PixelLM-style joint decoder (Ren et al., 2024b) along with the reasoning module. However, the results in Table J show significant degradation compared to our modular design. We attribute this to the *limited training data size*, which is insufficient for learning a strong segmentation decoder from scratch. The modular pipeline therefore provides better generalization and flexibility, especially when paired with large pretrained segmenters such as SAM2 (Ravi et al., 2024).

*Table J.* Comparison between end-to-end (E2E) joint decoder and our adopted modular design.

| Method | #Token↓ | gIoU (%)↑ | cIoU (%)↑ |
|---|---|---|---|
| E2E Joint Decoder | **40.15** | 59.52 | 56.65 |
| Ours (Modular) | 46.98 | **63.81** | **62.69** |

**About Efficiency Metrics.** Finally, we ablate the length penalty formulation in our proposed efficiency metrics. We take SAT as an example and compare it with CoT-Valve (Ma et al., 2025b), as illustrated in Table K. A direct division of accuracy by token length (Ma et al., 2025b) causes unstable behavior across different reasoning lengths. In contrast, the square-root scaling stabilizes performance, avoiding excessive penalization of informative yet slightly longer chains.

**Ablation on Penalty Weight.** To verify the robustness of our soft length penalty, we examine the influence of the penalty weight $\beta$ (in Eq. 2) on the validation set. As shown in Table L, our default choice ($\beta = 2 \times 10^{-3}$) yields the optimal performance. A smaller weight ($1 \times 10^{-3}$) provides insufficient constraints, resulting in longer token sequences (62.53) and suboptimal accuracy due to residual overthinking. Conversely, a larger weight ($4 \times 10^{-3}$) is overly aggressive, reducing the token count while harming segmentation performance by truncating necessary reasoning steps.

*Table L.* **Ablation on penalty weight.** Analyzing the impact of the soft penalty factor $\beta$.

| Penalty Weight ($\beta$) | #Token↓ | gIoU (%)↑ | cIoU (%)↑ |
|---|---|---|---|
| $1 \times 10^{-3}$ | 62.53 | 62.16 | 60.42 |
| $\mathbf{2 \times 10^{-3}}$ **(Ours)** | **46.98** | **63.81** | **62.69** |
| $4 \times 10^{-3}$ | 38.15 | 61.43 | 59.77 |

**Ablation on Penalty Function.** We investigate whether a non-linear penalty better models the cost-benefit curve of reasoning by comparing our linear design with a quadratic formulation: $s(L_{\text{used}}, L_{\text{budget}}) = 1 - \beta \cdot (L_{\text{used}} - L_{\text{budget}})^2$. As shown in Table M, while the quadratic penalty aggressively reduces token usage, it leads to a degradation in segmentation performance compared to our linear approach. This indicates that the quadratic penalty imposes an overly strict constraint that may truncate essential reasoning steps, confirming that the linear approximation offers a more stable and effective trade-off for policy optimization.

*Table M.* **Ablation on penalty function.** Comparison between linear and quadratic penalty formulations on the validation set.

| Penalty Function | #Token↓ | gIoU(%)↑ | cIoU(%)↑ |
|---|---|---|---|
| No Penalty | 90.79 | 61.63 | 52.56 |
| **Linear (Ours)** | 46.98 | **63.81** | **62.69** |
| Quadratic | **33.42** | 61.52 | 54.88 |

### C.2. Additional Qualitative Results

In Figure B, we provide additional qualitative comparisons between our PIXELTHINK and Seg-Zero, including both segmentation masks and reasoning chains for reference. Across a range of different scenarios, PIXELTHINK consistently

yields more accurate segmentation results while generating significantly shorter reasoning chains, highlighting its superior *efficiency* and *effectiveness*.

## C.3. Failure Cases and Analyses

**Failure Case Categorization.** To further validate the effectiveness of our difficulty decomposition and explore the limitations of our method, we conduct a manual analysis of 150 failure cases sampled from both the validation and test sets. Each failure was categorized into *one or more* sources of error. The results are summarized in Table N. We find that more than 84% of failures can be attributed to the three primary dimensions already modeled in our benchmark, including scene complexity, segmentation challenge, and linguistic ambiguity. These general findings support the validity of our benchmark design while also highlighting directions for future refinement.

**Comparative Analysis with Seg-Zero.** To quantify the impact of unconstrained overthinking on downstream performance, we isolate the failure modes inherently driven by the reasoning process and compare them against the Seg-Zero baseline. As shown in the bottom section of Table N, a significant portion (23.3%) of failures in Seg-Zero is directly caused by *Reasoning Chain Structural Errors*, where the model generates redundant or conflicting spatial descriptions that confuse the downstream mask decoder. By adaptively regulating the token budget based on task difficulty, PIXELTHINK explicitly suppresses these redundant structures, reducing such structural errors to 8.7%. Furthermore, failures stemming from *Vision-Language Grounding Mismatch* decrease from 15.3% to 10.7%. This indicates that concisely regularized reasoning chains provide cleaner, unambiguous textual priors for the segmentation head, thereby validating the regularization effect of our efficiency-aware reward design.

*Table N.* Failure case categorization on 150 samples. The bottom section isolates failure modes inherently driven by the reasoning process, where PIXELTHINK demonstrates significant reductions compared to the Seg-Zero baseline.

| Failure Category | Description | Seg-Zero | **PIXELTHINK** (Ours) |
|---|---|---|---|
| *General Failure Dimensions* | | | |
| Scene Complexity | Dense layouts, occlusions, background noise | – | 31.3% |
| Segmentation Challenge | Fine-grained boundary distinctions, spatial relations | – | 36.7% |
| Linguistic Ambiguity | Vague or multi-referent expressions | – | 18.7% |
| Annotation Ambiguities | Inherent label ambiguities or annotation inconsistencies | – | 4.7% |
| *Reasoning-Driven Failure Modes (Comparative Analysis)* | | | |
| Vision-Language Grounding Mismatch | Reasoning chain correct, but spatial token regression inaccurate | 15.3% | **10.7%** |
| Reasoning Chain Structural Errors | Redundant/conflicting reasoning steps affecting spatial grounding | 23.3% | **8.7%** |

**Visual Examples.** In Figure C, we present visual failure cases to illustrate limitations of the current approach. In the first example, both Seg-Zero and PIXELTHINK produce incomplete segmentation results, as multiple objects in the scene satisfy the referring conditions. In the second example, Seg-Zero fails during the reasoning process by misidentifying the target object, yet still generates a partially correct mask. In contrast, our method correctly identifies the target but yields an imperfect segmentation mask. These cases reveal a key limitation of the current decoupled architecture, where the reasoning outputs and final masks are not always well aligned. Table J further presents experiments using an end-to-end joint optimization scheme for the decoder; however, its performance remains limited due to the scarcity of training data. In the future, we will explore tighter integration and joint optimization with more training data to enhance consistency and overall performance.

## C.4. Generalizability to Next-Generation Architectures

The efficiency-aware reward scheme of PIXELTHINK features a strict modular design that decouples the reasoning MLLM from the mask decoder, enabling seamless integration with advanced, next-generation large vision-language models. While this architectural integration is straightforward, the reinforcement fine-tuning (RFT) process introduces inherent optimization complexities. Policy gradient methods like GRPO are highly sensitive to the base model's reference policy distribution, meaning that transitioning to a structurally updated backbone fundamentally shifts this initial distribution.

Consequently, applying RFT to new architectures requires a systematic recalibration of hyperparameters, such as the KL-divergence penalty, learning rate, and reward scaling, to ensure stable convergence and prevent reward hacking. To facilitate this adaptation process and support future research, we provide the complete codebase for newly released architectures (Bai et al., 2025a) on our project page.

# D. Additional Observations and Analyses

In this section, we provide complementary observations to further interpret the experimental results presented in the main paper. Specifically, we examine several empirical phenomena observed during training and evaluation: (1) the discrepancy between the token budget and the actual reasoning length, (2) the convergence of reasoning lengths across varying difficulty levels, and (3) the trade-off between concise reasoning and completeness in comparison to Seg-Zero (Liu et al., 2025b).

## D.1. Discrepancy between Token Budget and Reasoning Length

In our experiments, we observe that the model trained with the proposed reward framework frequently generates reasoning chains that are significantly shorter than the predefined token budget. This behavior can be attributed to several factors:

**Soft Length Penalty Encourages Conservative Generation.** Our reward function incorporates a *soft length penalty* that linearly penalizes the use of tokens exceeding the expected budget. Unlike hard truncation, this approach allows for flexibility while implicitly encouraging the model to stay within budget. Therefore, the model learns to *avoid unnecessary token usage* unless it contributes to improved task performance.

**Accuracy-dominant Reward Prevents Token Inflation.** The final reward integrates segmentation accuracy with alignment to the expected reasoning length. Since $\mathcal{R}_{\text{original}}$ primarily governs the reward dynamics, longer reasoning chains that do not lead to performance gains are *implicitly penalized*. This design encourages the generation of concise yet informative reasoning, where token usage is closely aligned with task utility.

## D.2. Convergence of Reasoning Length across Difficulty Levels

Although our training framework assigns distinct token budgets according to difficulty levels, we notice that the final reasoning lengths across all categories tend to converge within a relatively narrow range. Several determinants account for this phenomenon:

**Shared Decoder and Autoregressive Generation Bias.** The reasoning model employs a *unified decoder* with *same prompt* to generate reasoning chains for all samples. Since this decoder is optimized across tasks with varying levels of difficulty, it learns an averaged generation pattern and tends to favor a stable reasoning length distribution. This behavior is further reinforced by the autoregressive nature of decoding, through which the model implicitly learns a preferred stopping condition based on distributional patterns observed during training.

**Conservative Token Budget Design.** The token budget upper bounds for each difficulty group are set conservatively high to avoid premature truncation. The soft penalty is applied only when the reasoning length exceeds the budget and does not actively encourage the model to approach the upper limit. This design allows the model to naturally converge to a reasoning length below the threshold.

## D.3. `RScore` Performance Compared to Seg-Zero

While our method surpasses Seg-Zero in both segmentation accuracy and inference efficiency, we observe slightly lower values in the `RScore`, which evaluates the quality of the generated reasoning chains. This can be traced to limitations in the design of the `RScore` metric:

**`RScore` Emphasizes Completeness without Length Awareness.** RScore is computed based on three criteria: *completeness*, *grounding*, and *fluency*. Notably, the metric does not consider the brevity or efficiency of the generated reasoning. Thus, longer reasoning chains often receive higher completeness scores, even when parts of the explanation may be redundant.

**PIXELTHINK Prioritizes Efficiency and Accuracy.** PIXELTHINK is designed to generate concise yet informative reasoning under length-aware constraints. While our reasoning is more efficient, it may omit minor details which can lead to slightly lower completeness scores. However, these omissions *do not necessarily affect segmentation accuracy*, which remains higher in our approach.

**`RST` Facilitates a More Equitable Evaluation.** To address this limitation, we propose **Reasoning Score per Token** (`RST`), which normalizes `RScore` by both model size and the number of generated tokens. This metric offers a more holistic assessment of reasoning quality relative to computational cost, enabling fairer comparisons between models with varying reasoning lengths.

### D.4. Additional Background: Resource Rationality

Cognitive science research (Griffiths et al., 2015) has introduced the notion of resource rationality, which posits that intelligent agents optimize their performance given bounded cognitive resources (Lieder & Griffiths, 2020; Ma & Woodford, 2020). In this perspective, reasoning length or cognitive effort is not maximized unconditionally but adaptively allocated depending on task demands. Our formulation in PIXELTHINK is conceptually aligned with this principle: by regulating reasoning length based on task difficulty and uncertainty, the model achieves a more favorable trade-off between accuracy and efficiency. We include this discussion here to contextualize our approach within broader studies on bounded rationality and adaptive resource allocation in human cognition.

### D.5. Uncertainty-Driven Reward Hacking

As demonstrated in Table 4, combining external task difficulty and internal model uncertainty is critical for optimal performance. Relying exclusively on internal uncertainty $\mathcal{U}$, derived from token-level probability margins (Section 3.2), leaves the reinforcement learning policy vulnerable to reward hacking. Specifically, the model exploits the GRPO process by generating excessively verbose and repetitive reasoning chains. This prolonged output artificially inflates the cumulative uncertainty score, enabling the model to bypass length penalties and improperly expand its token budget.

This unconstrained generation destabilizes training by inducing high advantage variance within GRPO groups and introducing noisy, conflicting visual cues that degrade downstream segmentation accuracy. Anchoring the reward function with an objective, externally estimated task difficulty metric $\mathcal{D}$ prevents this degenerative behavior. By enforcing a strict token allowance for straightforward scenarios, the external difficulty prior suppresses unnecessary text expansion and regularizes the policy to produce concise, informative visual reasoning.

## E. Further Discussions

In this section, we further discuss the limitations of our work, highlight directions for future research and consider the potential societal impact.

### E.1. Limitation and Future Work

As the first attempt to enable efficient reasoning in reasoning segmentation, our method emphasizes simplicity and practicality, focusing on token-level control guided by task difficulty and model uncertainty. However, our design still relies on coarse-grained difficulty scores and manually defined token budget rules, which may limit adaptiveness in more complex scenarios. We also appreciate the concern regarding the reliance on a large external model for difficulty annotation. While this is a one-time offline cost, it introduces a dependency that future work could address by distilling this capability into smaller, more efficient scoring modules. Regarding the optimization objective, we currently employ a linear soft length penalty for training stability. Exploring more sophisticated non-linear penalty functions or learnable cost strategies represents a promising avenue to better model the trade-off between efficiency and accuracy.

Additionally, the reasoning and segmentation stages are loosely coupled. While we have demonstrated strong generalization across domains and model scales (*e.g.*, 3B and 7B), the applicability of our framework to distinct model architectures remains to be fully validated due to current infrastructure constraints. Extending our adaptive paradigm to these diverse model families is a priority for future research. In the future, we will explore more precise and robust difficulty estimation by leveraging self-supervised signals in combination with human feedback, as well as developing finer-grained, learnable token allocation strategies. Furthermore, integrating reasoning and segmentation into a joint optimization framework improves consistency and overall performance. Extending the proposed paradigm to other vision-language tasks such as visual commonsense reasoning (Feng et al., 2026; Zhang et al., 2026), video understanding and generation (Li et al., 2024; Fei et al., 2024; Kong et al., 2025) further demonstrates its generalizability and practical value.

### E.2. Potential Societal Impact

In this work, a reinforcement learning-based fine-tuning scheme is proposed for efficient reasoning in segmentation tasks, with potential applications in domains such as autonomous driving, robotics, and medical imaging. By enabling more efficient and interpretable visual reasoning, our method supports safer and more transparent decision-making in high-stakes scenarios. However, as with many vision-language models, our approach depends on large-scale pretrained models, which

may carry biases from their training data. Moreover, automated reasoning systems could be misapplied in contexts such as surveillance or critical decision-making without adequate human oversight. We advocate for responsible deployment and encourage further research on fairness, robustness, and transparency to ensure beneficial societal impact.

## (a) Easy Samples

**The pot lid**

"scene": 2
"seg": 4
"language": 2

**Reference Chain**
The pot lid is **white and positioned on top of the cup**, with a small knob at its peak. It is located directly above the cup's opening.

**The area that displays the time**

"scene": 2
"seg": 3
"language": 2

**Reference Chain**
The area that displays the time is **located at the top of the device, featuring red digital numbers showing "11:28 PM"**. It is positioned above the buttons and within a black rectangular display.

**The sauce**

"scene": 2
"seg": 4
"language": 1

**Reference Chain**
The sauce is the **bright red, chunky topping located in the center of the bowl**, covering the pasta. It contrasts sharply with the pale yellow noodles around it.

**The object that helps to keep the neck warm**

"scene": 4
"seg": 3
"language": 2

**Reference Chain**
The object is **a black scarf wrapped around the person's neck**. It is positioned just below the chin and extends down the front of the body.

## (b) Medium Samples

**The object that the dogs battle for**

"scene": 4
"seg": 6
"language": 3

**Reference Chain**
The object is **white with blue accents and is located in the mouths of both dogs**, positioned between them. It appears to be a toy they are tugging on.

**Something showing the price**

"scene": 3
"seg": 5
"language": 2

**Reference Chain**
Look for the **yellow sticker on the left side of the container**; it displays the price in black text. The price is clearly marked as "430".

**What object in the picture could provide shade and protection from the sun during the event?**

"scene": 4
"seg": 3
"language": 5

**Reference Chain**
Look for **the large white umbrella positioned behind the red sports car**; it provides shade and protection from the sun. It is located near the center-right of the image.

**Something that controls motion direction in games**

"scene": 4
"seg": 5
"language": 3

**Reference Chain**
The object is **located on the game's control panel**, near the bottom left side of the arcade machine. It is a **black joystick** positioned above the blue section of the panel.

## (c) Hard Samples

**The fishes whose mouth are open**

"scene": 8
"seg": 7
"language": 4

**Reference Chain**
**Step 1: Identify the fish with visible mouths:** In the image, there are numerous fish closely packed together. To determine which fish have their mouths open, focus on the area around the head of each fish where the mouth would be located. Look for any gaps or separations in the outline that suggest an opening.
**Step 2: Analyze the shape and position of the mouth:** Once potential candidates with open mouths are identified, examine the shape and position of these openings. An open mouth will typically appear as a dark, irregularly shaped gap, often wider at the top and tapering towards the bottom. This is distinct from closed mouths, which form a smooth line along the jaw.
**Step 3: Confirm by comparing with closed mouths:** To ensure accuracy, compare the identified open mouths with those that are clearly closed. Closed mouths will show no separation between the upper and lower jaws, forming a continuous line. By contrasting these features, it becomes easier to confirm which fish indeed have their mouths open, based on the visual evidence provided by the image's structure and details.

**If pigs are shot and killed, they usually fall to the ground. Which pigs in the picture are most likely to still be alive?**

"scene": 5
"seg": 7
"language": 8

**Reference Chain**
**Step 1: Assess the posture and position of the pigs.** The pigs in the image are standing upright within their individual stalls, which is a typical posture for live pigs. If a pig were shot and killed, it would likely collapse to the ground due to the loss of muscle control and support. The fact that these pigs are maintaining an upright stance suggests they are still alive.
**Step 2: Examine the environment and context.** The pigs are in a controlled environment, such as a farm or slaughterhouse, where they are kept in individual stalls. This setting implies that the pigs are being managed and monitored, and there is no visible evidence of recent shooting or killing activity. The absence of blood, signs of distress, or other indicators of death further supports the conclusion that the pigs are alive.
**Step 3: Consider the physical condition and behavior.** The pigs appear to be in good physical condition, with no visible injuries or signs of trauma. Their ears are perked up, and their bodies are not stiff or rigid, which would be expected if they were dead. Additionally, the pigs seem to be calmly standing in their stalls, indicating that they are conscious and responsive to their surroundings. These observations strongly suggest that the pigs are still alive.

*Figure A.* **Qualitative examples from ReasonSeg-DIFF.** Representative samples are shown with annotated difficulty scores and corresponding reference reasoning chains. For easy and medium cases, short reasoning chains are preferred to capture concise visual cues, whereas hard cases benefit from longer chains that reflect more elaborate reasoning over ambiguous or complex scenes. These examples illustrate the diverse reasoning requirements across difficulty levels and support more fine-grained evaluation.

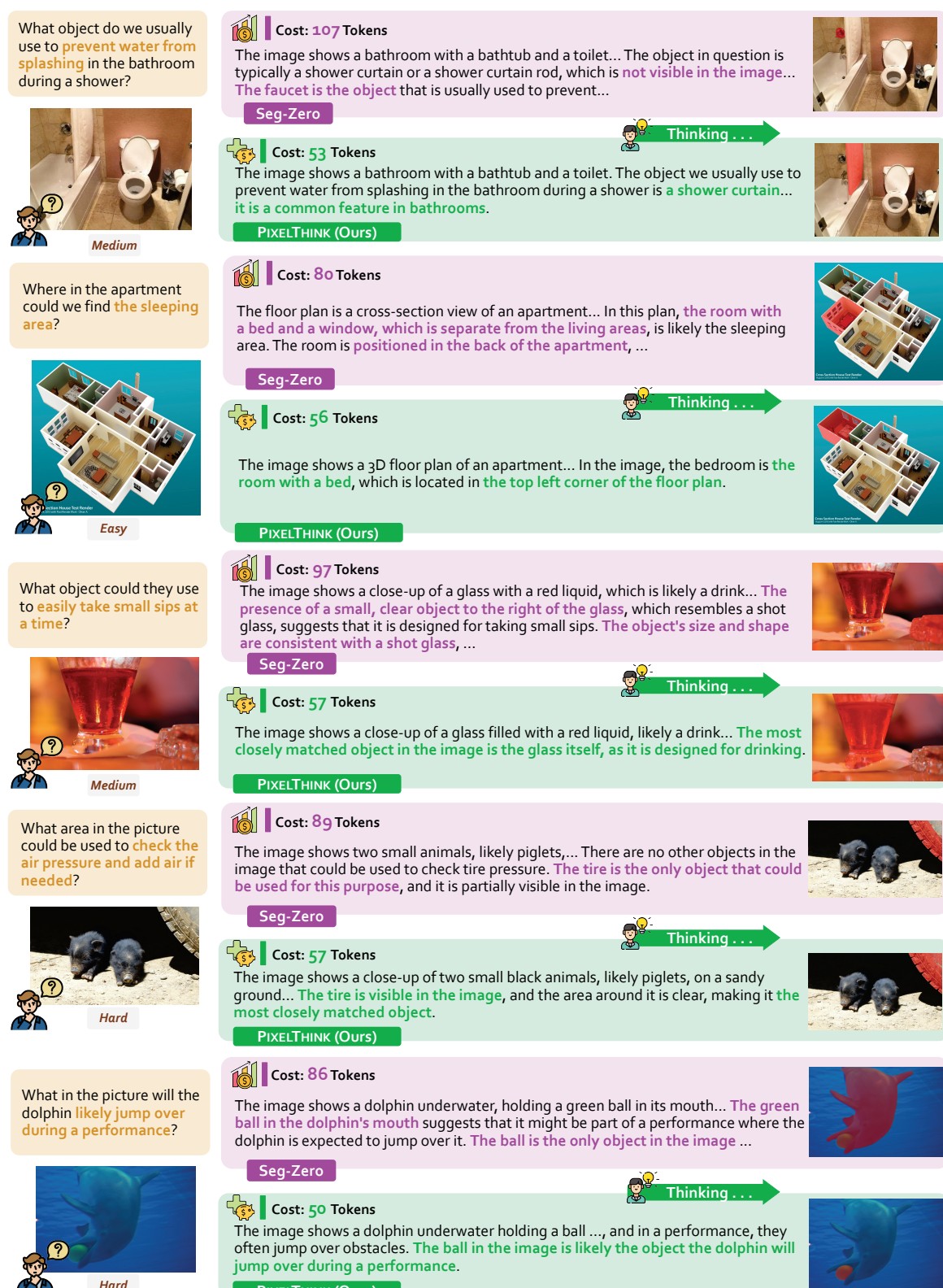

*Figure B.* Additional **qualitative comparisons** between Seg-Zero (Liu et al., 2025b) and the proposed PIXELTHINK, annotated with task difficulty levels (*Easy*, *Medium*, *Hard*). Our method demonstrates consistent improvements in segmentation accuracy across varying difficulty levels, accompanied by substantially shorter reasoning chains. These examples highlight the effectiveness of our efficient reasoning framework in mitigating *overthinking* while maintaining or improving segmentation quality.

The objects that can be used to **produce rhythmic sounds**.

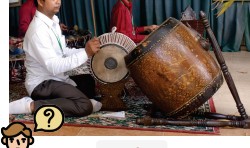

*Medium*

**Cost: 130 Tokens**

The image shows a group of individuals engaged in a musical performance... **The drum is positioned on a stand**, ..., which is a common method for producing rhythmic sounds... The other objects in the image, such as **the smaller drums and the individuals**, are secondary...

**Seg-Zero**

Thinking . . .

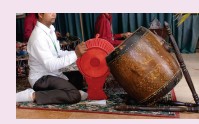

**Cost: 61 Tokens**

The image shows a group of people playing traditional drums. The most closely matched object is **the large drum in the foreground**, which is being played by a person in a white shirt. **The other drums in the background are also visible**, but the one in the foreground is the most prominent...

**PIXELTHINK (Ours)**

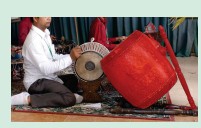

It is too hard for the man to lift up, can you segment **something that he should drop**?

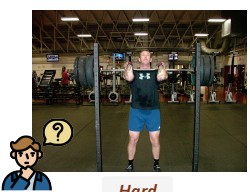

*Hard*

**Cost: 114 Tokens**

The image shows a man performing a weightlifting exercise, ... **There is no indication in the image that the weight is too heavy for the man to lift**, as he appears to be managing the lift with proper form. The question seems to be hypothetical or based on **a misunderstanding of the image content**.

**Seg-Zero**

Thinking . . .

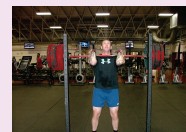

**Cost: 60 Tokens**

The image shows a man performing a weightlifting exercise. **The barbell he is lifting appears to be too heavy for him, as indicated by his strained expression and the position of the barbell**. The most closely matched object to drop is the barbell, which is the main object he is lifting.

**PIXELTHINK (Ours)**

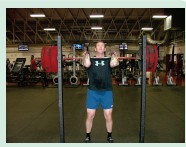

*Figure C.* **Failure cases** of Seg-Zero (Liu et al., 2025b) and the proposed PIXELTHINK. The first example illustrates ambiguity in the referring expression, resulting in incomplete segmentation masks from both methods. The second example reveals a mismatch between reasoning and segmentation: Seg-Zero fails to identify the correct object in its reasoning, while PIXELTHINK correctly interprets the instruction but produces a partially inaccurate mask.

# F. License and Consent with Public Resources

In this section, we outline the details of responsible release and acknowledge the use of public resources that supported this work.

## F.1. Responsible Release

Our work focuses on enhancing efficiency and controllability in reasoning segmentation using publicly available models and datasets. The proposed benchmark, ReasonSeg-DIFF, is constructed based on ReasonSeg (Lai et al., 2024), which includes human-annotated referring expressions and segmentation masks from natural scenes. We ensure that no private, sensitive, or copyrighted content is introduced during data processing. All models employed in our paper are open-sourced under appropriate licenses as detailed in the following subsection, and we do not release any newly trained large-scale models that could pose potential misuse risks. Upon releasing our benchmark and codebase, we will include a clear license, data usage policy, and guidelines to discourage applications involving sensitive attributes, surveillance, or unauthorized identification.

## F.2. Public Datasets

All experiments and the construction of our benchmark are conducted using the following publicly available datasets:

- ReasonSeg (Lai et al., 2024)[2] . . . . . . . . . . . . . . . . . . . . . . . . . . . . . . . . . . . . . . . . . . . . . . . . . . . . . . . . . . . . . . . . . . . . . . Apache License 2.0
- RefCOCO (+/g) (Yu et al., 2016)[3] . . . . . . . . . . . . . . . . . . . . . . . . . . . . . . . . . . . . . . . . . . . . . . . . . . . . . . . . . . . . . . Apache License 2.0
- MS COCO (Lin et al., 2014)[4] . . . . . . . . . . . . . . . . . . . . . . . . . . . . . . . . . . . . . . . . . . . . . . . . . . . . Other (specified in description)

## F.3. Public Models and Implementation

We compare and validate the effectiveness of the proposed method using the following publicly available models and source codes:

- Qwen2.5-VL (Bai et al., 2025b)[5] . . . . . . . . . . . . . . . . . . . . . . . . . . . . . . . . . . . . . . . . . . . . . . . . . . . . . . . . . . . . . Apache License 2.0
- SAM2 (Ravi et al., 2024)[6] . . . . . . . . . . . . . . . . . . . . . . . . . . . . . . . . . . . . . . . . . . . . . . . . . . . . . . . . . . . . . . . . . . . Apache License 2.0
- Seg-Zero (Liu et al., 2025b)[7] . . . . . . . . . . . . . . . . . . . . . . . . . . . . . . . . . . . . . . . . . . . . . . . . . . . . . . . . . . . . . . . . . Apache License 2.0
- verl (Sheng et al., 2024)[8] . . . . . . . . . . . . . . . . . . . . . . . . . . . . . . . . . . . . . . . . . . . . . . . . . . . . . . . . . . . . . . . . . . . . Apache License 2.0
- L1 (Bai et al., 2025b)[9] . . . . . . . . . . . . . . . . . . . . . . . . . . . . . . . . . . . . . . . . . . . . . . . . . . . . . . . . . . . . . . . . . . . . . . . . . . MIT License

---

[2]https://github.com/dvlab-research/LISA.
[3]https://github.com/lichengunc/refer.
[4]https://cocodataset.org.
[5]https://github.com/QwenLM/Qwen2.5-VL.
[6]https://github.com/facebookresearch/sam2.
[7]https://github.com/dvlab-research/Seg-Zero.
[8]https://github.com/volcengine/verl.
[9]https://github.com/cmu-l3/l1.

