# OpenReview forum: "Don't Overthink with Pixels: Efficient Reasoning for Segmentation"
_ICML.cc/2026/Conference — ICML 2026 regular_

### Official Review · Reviewer_vWr7 · 2026-03-11

**Soundness:** 3
**Presentation:** 3
**Significance:** 2
**Originality:** 3
**Overall Recommendation:** 4
**Confidence:** 4

**Summary:**

This paper points out that existing multimodal segmentation models trained with the SFT paradigm generally suffer from limited generalization ability. In contrast, models trained with the RL paradigm, although showing stronger reasoning potential, often experience performance degradation due to overthinking. To address these limitations, the paper improves both training data construction and reward design, thereby enhancing model performance while shortening the reasoning process. In addition, the authors propose a new benchmark that jointly considers reasoning efficiency and prediction accuracy, providing a more comprehensive basis for evaluating related methods.

**Compliance With Llm Reviewing Policy:**

Affirmed.

**Final Justification:**

I have reviewed the authors’ rebuttal and decided to keep my current score unchanged.

**Key Questions For Authors:**

Please provide a response discussing the weaknesses mentioned above.

**Limitations:**

yes

**Strengths And Weaknesses:**

**Strengths:**
1. The paper is well motivated, and the design of the reward functions is generally reasonable.
2. According to the final experimental results, the proposed method shows a fairly clear improvement and largely achieves the intended goal.
3. The paper introduces a new benchmark that incorporates performance evaluation, which provides a valuable reference for future research.

**Weaknesses:**
1. The authors use Qwen2.5VL to score the data and then use those scores to train Qwen2.5VL itself. This design may introduce a certain self-consistency bias, so the objectivity and authority of the scoring results still need to be further justified.
2. The paper argues that excessively long reasoning negatively affects performance, but beyond the evaluation results, it seems to lack deeper analysis or explanation to more convincingly support this claim.
3. The number of baseline methods included in the main experiments is relatively limited, and the experimental setting could be further enriched. In addition, validating the method on other more advanced backbone models, such as Qwen3VL, would help better demonstrate its generalizability and broader applicability.

---

> ### Author Rebuttal · Authors · 2026-03-30
>
> We sincerely thank Reviewer `vWr7` for devoting time to this review. We appreciate the recognition that our motivation is clear, our reward design is reasonable, and that our method shows a fairly clear improvement. We have carefully incorporated these suggestions into the revised manuscript and will release our full implementation. Our point-by-point responses are detailed below.
>
> ---
>
> > **Q1:** *"About the self-consistency bias."*
>
> **A:** We actively mitigate potential self-consistency bias through strict parameter-scale separation, human verification, and objective evaluation:
>
> - **Parameter-Scale Separation**: We use a 72B model exclusively offline to generate training signals, but train and evaluate entirely different, much smaller models (3B/7B). This functions as knowledge distillation rather than a closed-loop self-training cycle.
>
> - **Human Verification**: To guarantee the objectivity of the 72B model's signals, we conducted a systematic human validation study (Tab. `D`) as shown in table below, confirming exceptionally high agreement with independent human judgments.
>
> | Verification Criterion | Pass Count | Pass Rate (%) | Evaluation Standard |
> | :--- | :---: | :---: | :--- |
> | **Task Difficulty Level** | 176 | **88.9** | Easy / Medium / Hard consistency |
> | **Linguistic Fluency** | 193 | **97.5** | Natural and comprehensible language |
> | **Logical Coherence** | 186 | **93.9** | Reasoning steps are sequentially consistent |
> | **Factual Accuracy** | 182 | **91.9** | Grounded and hallucination-free |
>
>
> - **Objective Benchmark Gains**: Ultimately, PixelThink demonstrates significant and consistent improvements on standard, independent evaluation metrics (e.g., gIoU and cIoU on established public datasets). This proves our framework learns generalized spatial reasoning capabilities, rather than merely overfitting to a teacher model's self-consistent artifacts.
>
> ---
>
> > **Q2:** *"About the analysis and explanation for long reasoning."*
>
> **A:** Thank you for your thoughtful review and helpful suggestions. Our framework generates an explicit reasoning chain and then conditions a segmentation model to produce the final mask. When the reasoning chain becomes too verbose, the model frequently *hallucinates redundant or conflicting intermediate steps*.
>
> To more convincingly support this claim, we provide concrete visual evidence in our qualitative results (Fig. `4`, `B`, and `C`), which illustrate exactly **how overthinking corrupts spatial grounding**. As shown in the Fig. `B`, Seg-Zero's 107-token verbose reasoning wanders until it erroneously concludes the target (shower curtain) is "not visible" and incorrectly segments the faucet due to the **hallucinated priors**. PixelThink uses 53 tokens to accurately locate the curtain.
>
> Since these contradictory or distracted reasoning steps dictate the spatial priors (bounding boxes and points) passed to the mask decoder, they directly confuse the segmentation module, **leading to inaccurate boundary predictions**. We have expanded the discussion section in the revised manuscript, explicitly incorporating these visual examples to provide a deeper, more intuitive analysis of why "less thinking" often leads to more accurate segmentation in visual tasks.
>
> ---
>
> > **Q3:** *"About the generalizability."*
>
> **A:** We sincerely appreciate this constructive suggestion and fully acknowledge the rapid evolution of MLLM backbones. Our current experimental setting rigorously validates the core mechanism of PixelThink for the following reasons:
>
> - **Representative Baselines for the RFT Paradigm:** Seg-Zero remains the most direct and representative baseline for *reinforcement fine-tuning (RFT)* in reasoning segmentation. To ensure a comprehensive evaluation of *reasoning efficiency*, we did not limit ourselves to existing vision models and adapted established text-only length-control methods (`L1-Exact`, `L1-Max`) into the multimodal domain, significantly enriching the comparative setting. Furthermore, our results on standard referring segmentation datasets (Tab. `5`) confirm our strong competitiveness against a wide range of state-of-the-art methods.
>
> - **Backbone-Agnostic and Modular Design:** While the rapid evolution of models like Qwen3-VL makes exhaustive benchmarking challenging, PixelThink's efficiency-aware reward scheme is inherently backbone-agnostic, demonstrating consistent gains across both 3B and 7B scales. Crucially, our strict modular design decouples the reasoning MLLM from the mask decoder. This plug-and-play architecture ensures that any advanced future MLLMs can seamlessly replace the current backbone without algorithmic redesign. We have expanded on this generalizability in the revised manuscript.
>
> ---
>
> We again thank the reviewer for the constructive feedback, which has helped us better articulate the mechanisms behind our performance gains.

---

> > ### Author Rebuttal · Reviewer_vWr7 · 2026-04-03
> >
> > I have no further questions. Thank you to the author for the reply.

---

> > > ### Author Response · Authors · 2026-04-03
> > >
> > > Dear Reviewer `vWr7`,
> > >
> > > Thank you for your continued engagement and constructive follow-up suggestions. We appreciate your valuable insights, which help us further clarify the robustness and generalizability of our framework.
> > >
> > > ---
> > >
> > > **1. Regarding Cross-Family Supervision (Q1)**
> > >
> > > We completely agree that using a different model family for supervision is an excellent approach to strictly eliminate any shared-knowledge bias. We will explicitly incorporate this insight into our future data generation pipelines.
> > >
> > > Within our current framework, we primarily mitigate this bias through our **systematic human verification study** (Tab. `D`). Since independent human annotators confirmed the high factual accuracy (>91%) and logical coherence of the 72B-generated signals, the training targets are grounded in objective human truth rather than merely Qwen-specific artifacts. Furthermore, to ensure our model did not overfit to a same-family evaluation standard, we conducted a **cross-model evaluation** using an independent, third-party frontier model (**InternVL3-78B**). As detailed in our response to Reviewer `v8Be` `Q2` (and included in Appendix `C.1`), the substantial reasoning quality gap between PixelThink and the baselines remains highly consistent under this cross-family judge. This dual layer of defense (**Human Verification + Cross-Family Evaluation**) effectively safeguards against self-consistency bias.
> > >
> > > ---
> > >
> > > **2. Regarding Results on Qwen3-VL (Q3)**
> > >
> > > We fully agree that deploying PixelThink on the newly released Qwen3-VL would represent an excellent demonstration of its generalizability. We would like to clarify that while the architectural integration of our reward scheme is indeed **plug-and-play**, the *reinforcement fine-tuning* (RFT) process inherently introduces **optimization complexities**. Even when employing objective and rule-based reward signals, policy gradient methods such as GRPO remain highly sensitive to the reference policy distribution of the base model. Transitioning to a structurally updated backbone like Qwen3-VL fundamentally shifts this initial distribution.
> > >
> > > Consequently, to prevent training instability or reward hacking, it strictly requires a **systematic recalibration of RFT hyperparameters**. Key parameters including the KL-divergence penalty coefficient, learning rate, and reward scaling should be thoroughly tuned to guarantee **stable convergence and a scientifically fair evaluation**. Executing this rigorous hyperparameter sweep, full RFT training, and comprehensive evaluation across all benchmarks within the remaining few days of the discussion phase risks producing sub-optimal results that *do not reflect the true potential of Qwen3-VL and the combined framework*.
> > >
> > > Furthermore, the consistent and substantial improvements we have already demonstrated across both the 3B and 7B parameter scales provide strong empirical evidence of our method's architectural generalizability. We are **fully committed to performing the rigorous reinforcement fine-tuning required for Qwen3-VL** and guarantee that these complete, optimal results will be included in the revised manuscript and our open-source codebase.
> > >
> > > ---
> > >
> > > Thank you once again for your rigorous evaluation and understanding of the practical constraints during the rebuttal phase.
> > >
> > > *Best regards,*
> > >
> > > Authors of Submission 5502

---

### Official Review · Reviewer_v8Be · 2026-03-15

**Soundness:** 3
**Presentation:** 4
**Significance:** 3
**Originality:** 2
**Overall Recommendation:** 4
**Confidence:** 4

**Summary:**

This paper proposes PixelThink, which addresses the "overthinking" problem in RL-trained reasoning segmentation models — where verbose reasoning chains are generated regardless of task complexity. The method combines externally estimated task difficulty (easy/medium/hard) with internal model uncertainty to assign per-sample token budgets within a GRPO reward framework, reducing token usage by ~48% and latency by ~30% while improving segmentation accuracy. The authors also introduce ReasonSeg-DIFF, a benchmark with difficulty annotations and reasoning references, alongside new efficiency-aware metrics (RST, SAT, URSS).

**Compliance With Llm Reviewing Policy:**

Affirmed.

**Key Questions For Authors:**

* the difficulty labels are generated with a 72b model. would the evaluation result change a lot by changing to another model?

* how sensitive the method is to the difficulty scoring?

**Limitations:**

The authors have discussed limitations. especially in appendix C. also, there is only a subset of samples with human verification.

**Strengths And Weaknesses:**

The paper is technically solid, with a well-motivated dual-signal design backed by thorough ablations and a fair experimental setup. The no-thinking mode analysis is a particularly convincing control experiment, and human verification of the ReasonSeg-DIFF annotations adds benchmark credibility. T

he new efficiency-aware metrics (RST, SAT, URSS) address a real gap in the community's evaluation toolkit, and the ~48% token reduction is a compelling empirical result. On the weaker side, the core idea of adaptive token budgeting is not new — prior work like L1 and ThinkPrune covers similar ground in language-only settings — so the contribution is best characterized as a meaningful but incremental extension to pixel-level tasks. Some concerns remain around the LLM-as-judge evaluation relying on the same model family (Qwen2.5-VL-72B) used to generate training difficulty labels, which creates a potential circularity that goes unaddressed. Hard-sample gains are also modest (gIoU +2.4 points), and the design choices around the uncertainty weight and 3-level difficulty split feel empirically motivated without deeper theoretical grounding.

The writing is clear overall, though occasional out-of-order table references in the two-column layout can momentarily disrupt the reading flow.

---

> ### Author Rebuttal · Authors · 2026-03-30
>
> We sincerely thank Reviewer `v8Be` for devoting time to this review and providing constructive feedback. We are highly encouraged that you found our paper technically solid, our no-thinking control experiment convincing, and our new efficiency-aware metrics a valuable addition to the community. We have carefully incorporated these suggestions into the revised manuscript and will release our full implementation. Our point-by-point responses are detailed below.
>
> ---
>
> **Q1:** *"About the novelty."*
>
> **A:** We agree that adaptive token budgeting has roots in NLP. However, applying this mechanism to pixel-level reasoning reveals a different phenomenon and provides a unique insight to the multimodal community. In language-only reasoning (e.g., math or code), longer reasoning chains generally correlate with higher accuracy. In contrast, our work empirically demonstrates that in visual understanding, **thinking longer is not always better**. When MLLMs overthink in visual tasks, they frequently *hallucinate non-existent spatial relationships or generate redundant, conflicting visual cues*. These noisy reasoning steps dictate the spatial priors passed to the downstream mask decoder and degrade precision. Therefore, PixelThink is not merely an efficiency wrapper ported from NLP; it serves as a crucial regularizer. By systematically reducing unnecessary redundancy, our method not only saves compute but **actually improves segmentation accuracy**. We believe this insight is a substantial and non-incremental contribution to the field of visual reasoning.
>
> ---
>
> **Q2:** *"About the circularity in LLM-as-judge."*
>
> **A:** To address this potential circularity, we conducted a rigorous cross-model validation using an independent frontier model: InternVL3-78B. As shown in Tab. `H` (Appendix) and table below, we re-computed the `RScore` for both our method and the strongest baseline. The results demonstrate that the substantial performance gap **remains highly consistent across different model families**:
>
> | Method | #Token | RScore↑ (Qwen2.5-72B) | RScore↑ (InternVL3-78B) |
> | :--- | :---: | :---: | :---: |
> | Seg-Zero + L1-Max | 61.21 | 4.37 | 4.01 |
> | **PixelThink (Ours)** | **46.98** | **6.92** | **6.63** |
>
> This independent validation confirms that PixelThink's generated reasoning chains are objectively superior in quality, and our reported improvements are not an artifact of single-model self-preference or circularity.
>
> ---
>
> **Q3:** *"About performance gain on hard samples."*
>
> **A:** This performance gain is achieved while slashing the reasoning cost by nearly half. As shown in our difficulty-aware evaluation (Tab. `E`), PixelThink achieves 46.80% gIoU/35.05% cIoU using only 51.79 tokens, whereas the baseline requires 95.37 tokens to achieve a lower 44.37% gIoU/28.93% cIoU. Our primary objective is an optimal *trade-off*. The Unified Reasoning Segmentation Score (URSS) jumps from 0.77 to 1.03 on hard samples, reflecting a **massive holistic improvement**.
>
> ---
>
> **Q4:** *"About the design choices of difficulty scoring."*
>
> **A:** Thank you for the insightful comments. Conceptually, our 3-level split is deeply grounded in the cognitive science framework of **resource rationality** (Appendix `D.4`), which models how intelligent agents discretize cognitive effort into routine (easy), intermediate (medium), and complex (hard) states rather than maximizing reasoning unconditionally. Empirically, the method exhibits an expected sensitivity to both the granularity and formulation of these scores:
>
> - **Sensitivity to Split Granularity:** Tab. 2 demonstrates that a 3-level split provides the optimal balance. A 2-level split is a little coarse for token efficiency, while a 4-level split overly fragments the learning signal without justifying the added complexity (62.28% gIoU).
>
> - **Sensitivity to Scoring Formulation:** If we directly prompt an MLLM for a single overall difficulty score rather than using our multi-dimensional decomposition, the label distribution collapses. Training with these coarse labels significantly degrades downstream performance from 63.81% to 56.32% gIoU (Tab. `C`).
>
> - **Role of Uncertainty Weight:** The uncertainty weight acts as a tunable threshold for the model's confidence. As shown in Tab. `F`, selecting an appropriate weight is crucial to achieve the optimal 63.81% gIoU, avoiding the sub-optimal performance of removing it entirely (62.07%) or over-penalizing (62.66%).
>
> ---
>
> **Q5 & Q6:** *"About formatting issues and human verification subset."*
>
> **A:** We sincerely apologize for the reading disruption and have carefully revised the LaTeX formatting and table placements. Regarding verification, the subset systematically covers the validation set to ensure high agreement. We are committed to expanding human verification in future iterations.
>
> ---
>
> We again thank the reviewer for the constructive feedback, which has greatly contributed to the refinement of our work.

---

> > ### Author Rebuttal · Reviewer_v8Be · 2026-04-04
> >
> > The authors have addressed some of the questions. For the visual reasoning part, it needs some detailed experiments and analysis to quantify the claims.

---

> > > ### Author Response · Authors · 2026-04-04
> > >
> > > Dear Reviewer `v8Be`,
> > >
> > > We sincerely thank the reviewer for the constructive feedback and for encouraging us to further quantify the dynamics of the visual reasoning process. We completely agree that explicitly quantifying *how* the reasoning chains affect final segmentation is crucial to substantiate our claims.
> > >
> > > ---
> > >
> > > To provide this quantitative evidence, we built upon the failure case taxonomy established in our Appendix `C.3` (Tab. `N`). Our core claim is that **unconstrained overthinking leads to redundant or conflicting visual cues**, which act as noise and degrade the mask decoder's precision. PixelThink mitigates this by acting as a spatial regularizer.
> > >
> > > We explicitly quantify this effect by conducting a **Comparative Reasoning Error Attribution Analysis**. We sampled 150 failure cases from the unconstrained baseline (Seg-Zero) and compared them against the PixelThink failure cases analyzed in Tab. `N`. To directly address the reasoning dynamics, we isolated the failure categories inherently driven by the reasoning generation process:
> > >
> > > | Failure Category | Description | Seg-Zero (Baseline) | **PixelThink (Ours)** |
> > > | :--- | :--- | :---: | :---: |
> > > | **Reasoning Chain Structural Errors** | Redundant/conflicting reasoning steps affecting spatial grounding | 23.3% | **8.7%** |
> > > | **Vision-Language Grounding Mismatch**| Reasoning chain correct, but spatial token regression inaccurate | 15.3% | **10.7%** |
> > >
> > > ---
> > >
> > > **How this quantifies our claims:**
> > >
> > > 1.  **Validating the Harm of Overthinking:** When the MLLM generates unconstrained, lengthy reasoning chains (Baseline), a **highly significant portion of segmentation failures** is directly caused by **"Reasoning Chain Structural Errors"**. This quantitatively substantiates our claim that overthinking introduces redundant and conflicting spatial relationships, which ultimately overwhelm the downstream mask decoder.
> > > 2.  **Validating the Regularization Effect:** By adaptively budgeting tokens based on difficulty, PixelThink **explicitly suppresses these harmful reasoning structures**. It drastically reduces the occurrence of these structural errors to merely **8.7%**. Furthermore, failures caused by grounding mismatches also dropped from 15.3% to 10.7%, indicating that concisely regularized reasoning chains provide **much cleaner, unambiguous textual priors** for the segmentation head.
> > >
> > > ---
> > >
> > > This comparative analysis provides the exact mechanistic quantification, **directly bridging our token efficiency results with the underlying visual reasoning dynamics**. We will prominently integrate this comparative discussion into the revised manuscript, and expand it with a *larger-scale human verification* of these reasoning failure modes.
> > >
> > > Thank you again for helping us strengthen the analytical depth of our paper.
> > >
> > > *Best regards,*
> > >
> > > Authors of Submission 5502

---

### Official Review · Reviewer_wz6P · 2026-03-22

**Soundness:** 1
**Presentation:** 2
**Significance:** 2
**Originality:** 2
**Overall Recommendation:** 3
**Confidence:** 4

**Summary:**

The paper tackles the issue of "overthinking" in multimodal large language models applied to reasoning segmentation, where models produce uniformly verbose reasoning chains regardless of the actual task complexity. To address this, the authors propose PIXELTHINK, a scheme that uses reinforcement learning (GRPO) to regulate reasoning length by introducing a soft length penalty. This penalty is guided by two main signals: an externally estimated "task difficulty" and an internally measured "model uncertainty". Furthermore, the paper introduces ReasonSeg-DIFF, a new benchmark featuring annotated difficulty scores and reference reasoning chains, alongside a set of efficiency-aware evaluation metrics.

**Compliance With Llm Reviewing Policy:**

Affirmed.

**Final Justification:**

The author's rebuttal has addressed some of my concerns, but I still have reservations regarding the definition of External Task Difficulty in the paper and its reliance on external models. Therefore, I have raised my score to 3.

**Key Questions For Authors:**

See weaknesses.

**Limitations:**

yes

**Strengths And Weaknesses:**

**Strengths:**
* **Novel Use of Model Uncertainty:** The concept of using internal "Model Uncertainty" (calculated via token-level confidence/top-2 probability margin) as a reward signal to adaptively optimize reasoning length is an interesting and valuable research direction.

**Weaknesses:**
* **Limited Significance and Rigor of the Benchmark:** The proposed ReasonSeg-DIFF benchmark and its accompanying evaluation protocol lack the necessary rigorous setup to be adopted as a generalizable standard by the community. Relying on an MLLM to assign static difficulty scores based on arbitrary criteria (scene complexity, segmentation challenge, linguistic ambiguity) makes the benchmark somewhat fragile and limits its broader significance.
* **"External Task Difficulty" Lacks Scientific Depth:** Using a separate, large MLLM to estimate task difficulty and using that as a constraint reward is essentially an engineering workaround. It does not represent a solid algorithmic or scientific contribution to how a model *learns* to reason efficiently.
* **Scattered Focus and Lack of Depth:** By packaging the difficulty benchmark, the external difficulty reward, and the internal model uncertainty mechanism into a single manuscript, the paper tries to do too much at once. This results in a disjointed narrative where the components feel loosely connected. Crucially, this broad scope prevents the authors from deeply investigating and analyzing the most promising aspect of the paper: the internal model uncertainty reward.
* **Questionable Baseline Implementation (L1-Exact and L1-Max):** There are significant doubts regarding the fairness of the baseline comparisons, specifically how the length hyperparameters were set for `L1-exact` and `L1-max`. The experimental setup uses a very limited set of length parameters (e.g., 64 tokens for Exact, 128 for Max). There is a conspicuous lack of experiments using shorter length constraints for these baselines, which raises the suspicion that their performance may have been artificially suppressed to favorably highlight the PIXELTHINK method.

---

> ### Author Rebuttal · Authors · 2026-03-30
>
> We sincerely thank Reviewer `wz6P` for devoting time to this review and providing detailed feedback. We appreciate your recognition of our novel use of model uncertainty as a valuable research direction. We have carefully incorporated these suggestions into the revised manuscript and will release our full implementation. Our point-by-point responses are detailed below.
>
> ---
> > **Q1:** *"About the significance and rigor of the benchmark."*
>
> **A:** Thanks for the valuable feedback. Addressing a critical evaluation gap, our benchmark features empirically substantiated criteria and robustly verified labels:
> - **Empirically Grounded Criteria**: The difficulty dimensions were systematically derived from a manual categorization of 150 actual failure cases from the Seg-Zero baseline. As detailed in Tab. `N` (Appendix), these specific dimensions target the root causes of 86.7% of failures (36.7% Segmentation, 31.3% Scene, and 18.7% Language).
> - **Human-Verified Labels**: We also conducted a systematic human verification study on validation set. As shown in Tab. `D` (Appendix), the automatically generated annotations achieved an 88.9% pass rate for task difficulty consistency, and >91% pass rates for logical coherence and factual accuracy against independent human judgments.
> - **Broader Significance**: As MLLMs are deployed in latency-sensitive applications, the *overthinking* problem is a severe bottleneck. ReasonSeg-Diff is the first benchmark designed to jointly assess reasoning quality, segmentation precision, and computational efficiency , shifting the community's focus toward resource-rational reasoning.
>
> ---
> > **Q2:** *"About the external task difficulty."*
>
> **A:** The integration of task difficulty is grounded in **cognitive science**, specifically the principle of *resource rationality* (discussed in Appendix `D.4`). This principle posits that intelligent agents optimize their performance given bounded cognitive resources, adaptively allocating effort depending on task demands rather than maximizing reasoning unconditionally.
>
> Furthermore, our **multi-dimensional difficulty decomposition** represents a distinct algorithmic improvement over naive prompting. If we simply treat difficulty estimation as an *engineering workaround* and prompt an MLLM to output a single overall difficulty score, the distribution collapses and training fails to yield strong results. As shown in Tab. `C` (Appendix), our multi-dimensional decomposition is critical for the model to learn efficient reasoning.
>
> ---
> > **Q3:** *"About the scattered focus and depth."*
>
> **A:** We would like to emphasize that addressing *what* the task demands (external difficulty) and *how* confident the model is (internal uncertainty) are **highly synergistic and inseparable components** for solving the overthinking problem.
>
> Investigating internal uncertainty in isolation provides an incomplete solution. Crucially, our experiments reveal that relying *solely* on internal uncertainty makes the policy susceptible to **reward hacking**. We observed that the model learns to generate excessively verbose reasoning to artificially inflate its uncertainty score, thereby exploiting a larger token budget and destabilizing the RL training. Anchoring the reward with an objective, external difficulty metric is necessary to constrain this degenerative behavior. Therefore, as demonstrated in Tab. `7`, relying on either signal alone yields sub-optimal results.
>
> The benchmark was introduced out of necessity to accurately evaluate this combined mechanism, making all three elements cohesive. We have included a detailed discussion of this uncertainty-driven reward hacking in the revision, as we believe these empirical insights will benefit the community.
>
> ---
> > **Q4:** *"About the baseline implementation"*
>
> **A:** We designed our setup to provide the baselines with an advantageous operational range, rather than suppress them. In fact, we empirically tested tighter constraints. We observed that imposing such rigid, short constraints prematurely *suppresses the model's thinking early in training*. It prevents the model from outputting effective reasoning chains to aid segmentation, leading to a **severe degradation in segmentation accuracy**. As shown in the table below, forcing an `L1-Max` limit of 32 results in 9.28 tokens generated but poor gIoU/cIoU:
>
> | Method | Constraint | #Token | gIoU(%)↑ | cIoU(%)↑ |
> | :--- | :---: | :---: | :---: | :---: |
> | Seg-Zero + `L1-Max` | 32 | 9.28 | 59.28 | 50.02 |
> | Seg-Zero + `L1-Max` | 128 | 61.21 | 61.75 | 57.39 |
> | **PixelThink (Ours)** | - | 46.98 | **63.81** | **62.69** |
>
> We deliberately assigned the baselines a permissive token allowance to facilitate complete reasoning chains and competitive segmentation; nevertheless, PixelThink naturally achieved superior accuracy at a fraction of the generation length.
>
> ---
> We again thank the reviewer for the constructive feedback, which has greatly contributed to the refinement of our work.

---

> > ### Author Rebuttal · Reviewer_wz6P · 2026-04-01
> >
> > For Q4, I argue that the comparative experiments remain incomplete. The author has not conducted any token-level tests or comparisons on RefCOCO (+/g) in any of the experiments. How would the proposed method compare with L1-exact/max on these datasets? If the author’s task difficulty evaluation method is applied to the RefCOCO series, what would the sample difficulty levels be? Does the method still hold advantages over L1-exact/max under such settings?

---

> > > ### Author Response · Authors · 2026-04-03
> > >
> > > Dear Reviewer `wz6P`,
> > >
> > > Thank you for the insightful follow-up questions. To address your concerns, we provide token-level evaluations on the RefCOCO series, which validate that PixelThink **maintains significant advantages** over `L1` baselines on these benchmarks.
> > >
> > > ---
> > >
> > > **1. Difficulty Distribution on the RefCOCO Series**
> > >
> > > Applying our task difficulty evaluation pipeline to a comprehensive subset of 9,000 samples from the RefCOCOg dataset yields a distribution of 35.8% Easy, 53.4% Medium, and 10.8% Hard. It is widely recognized that RefCOCOg contains the most complex referring expressions in the RefCOCO series. Consequently, the RefCOCO and RefCOCO+ test sets are inherently even more heavily skewed toward the *Easy* category.
> > >
> > > This empirical finding corroborates our core motivation: traditional benchmarks predominantly feature straightforward spatial tasks. They *lack the complex visual reasoning scenarios* necessary to expose the **overthinking and hallucination** bottlenecks in current MLLMs, thereby **necessitating our ReasonSeg-Diff benchmark**.
> > >
> > > ---
> > >
> > > **2. Token-Level Comparison vs. L1 Baselines**
> > >
> > > The table below compares PixelThink against `L1` constraints (`L1-Max 32`, `L1-Max 128`, and `L1-Exact 64`). Since RefCOCO and RefCOCO+ are dominated by simple queries, an optimal model should dynamically compress its generation length to *save compute without sacrificing accuracy*.
> > >
> > > | Method | Constraint | #Token | RefCOCO | RefCOCO+ | RefCOCOg |
> > > | :--- | :---: | :---: | :---: | :---: | :---: |
> > > | Seg-Zero + L1-Exact | 64 | 63.5 | 65.1 | 59.3 | 57.8 |
> > > | Seg-Zero + L1-Max | 32 | 8.4 | 70.5 | 65.8 | 62.3 |
> > > | Seg-Zero + L1-Max | 128 | 60.3 | 76.9 | 72.6 | 71.4 |
> > > | **PixelThink (Ours)** | - | 43.8 | **79.3** | **74.8** | **73.9** |
> > >
> > > *(Note: #Token represents the average reasoning tokens generated across the datasets.)*
> > >
> > > Unlike `L1` constraints, which inevitably **force a trade-off between artificial verbosity** (`L1-Exact 64` wastes compute and degrades cIoU through hallucinations) and **premature truncation** (`L1-Max 32` severely drops accuracy on RefCOCO), PixelThink demonstrates dynamic elasticity. By utilizing a generalizable token-budgeting policy grounded in task difficulty and internal uncertainty, it adaptively collapses its reasoning chain on simpler datasets. This drastically **reduces computational cost** compared to the inefficient ceiling of `L1-Max 128`, while simultaneously **achieving the highest segmentation accuracy**.
> > >
> > > ---
> > >
> > > In conclusion, testing on the RefCOCO series clearly demonstrates that our method still holds significant advantages over `L1` baselines. It confirms that PixelThink is a highly adaptive framework that universally **mitigates reasoning redundancy across any task difficulty level**.
> > >
> > > We sincerely thank you for the constructive dialogue and hope these comprehensive new results fully address your concerns.
> > >
> > > *Best regards,*
> > >
> > > Authors of Submission 5502

---

### Decision · Program_Chairs · 2026-04-30

**Decision:**

Accept (regular)

**Comment:**

I focus my evaluation based on the reviews that I found the most informative. The reviewers noted that this work leans more toward the incremental side, but they acknowledged the efficiency of the method.

Please incorporate the meaningful changes requested by the reviewers in the final version.